# Beyond Pinball Loss: Quantile Methods for Calibrated Uncertainty Quantification

**Youngseog Chung**
Machine Learning Department
Carnegie Mellon University
Pittsburgh, PA 15213
youngsec@cs.cmu.edu

**Willie Neiswanger**
Department of Computer Science
Stanford University
Stanford, CA 94305
neiswanger@cs.stanford.edu

**Ian Char**
Machine Learning Department
Carnegie Mellon University
Pittsburgh, PA 15213
ichar@cs.cmu.edu

**Jeff Schneider**
Robotics Institute
Carnegie Mellon University
Pittsburgh, PA 15213
schneide@cs.cmu.edu

## Abstract

Among the many ways of quantifying uncertainty in a regression setting, specifying the full quantile function is attractive, as quantiles are amenable to interpretation and evaluation. A model that predicts the true conditional quantiles for each input, at all quantile levels, presents a correct and efficient representation of the underlying uncertainty. To achieve this, many current quantile-based methods focus on optimizing the *pinball loss*. However, this loss restricts the scope of applicable regression models, limits the ability to target many desirable properties (e.g. calibration, sharpness, centered intervals), and may produce poor conditional quantiles. In this work, we develop new quantile methods that address these shortcomings. In particular, we propose methods that can apply to any class of regression model, select an explicit balance between calibration and sharpness, optimize for calibration of centered intervals, and produce more accurate conditional quantiles. We provide a thorough experimental evaluation of our methods, which includes a high dimensional uncertainty quantification task in nuclear fusion.

## 1 Introduction

Uncertainty quantification (UQ) in machine learning typically refers to the task of quantifying the confidence of a given prediction. This measure of certainty can be crucial in a variety of downstream applications, including Bayesian optimization [36, 45, 54], model-based reinforcement learning [64, 42, 10, 18], and in high-stakes predictions where mistakes incur large costs [52, 61].

While the common goal of UQ is to describe predictive distributions over outputs for given inputs, the representation of the distributional prediction varies across methods. For example, some methods assume a parametric distribution and return parameter estimates [65, 14, 37], while others return density function estimates, as is common in Bayesian methods [41, 38, 24, 3, 33, 50]. Alternatively, many methods represent predictive uncertainty with quantile estimates [15, 53, 58, 49, 27].

Quantiles provide an attractive representation for uncertainty because they can be used to model complex distributions without parametric assumptions, are interpretable with units in the target output space, allow for easy construction of prediction intervals, and can be used to efficiently sample from the predictive distribution via inverse transform sampling [22, 32]. Learning the quantile for a single

---

Code is available at https://github.com/YoungseogChung/calibrated-quantile-uq.

35th Conference on Neural Information Processing Systems (NeurIPS 2021).

quantile level is a well studied problem in quantile regression (QR) [30, 32], which typically involves optimizing the so-called *pinball loss*, a tilted transformation of the absolute value function. Given a target $y$, a prediction $\hat{y}$, and quantile level $\tau \in (0, 1)$, the pinball loss $\rho_\tau$ is defined as

$$\rho_\tau(y, \hat{y}) = (\hat{y} - y)(\mathbb{I}\{y \leq \hat{y}\} - \tau). \tag{1}$$

By training for all quantiles simultaneously, recent works have made concrete steps in incorporating QR methods to form competitive UQ methods which output the full predictive distribution [51, 58].

In this work, we highlight some limitations of the pinball loss and propose several methods to address these shortcomings. Specifically, we explore the following:

- **Model agnostic QR.** Optimizing the pinball loss often restricts the choice of model family for which we can provide UQ. We propose an algorithm to learn all quantiles simultaneously by utilizing methods from conditional density estimation. This algorithm is agnostic to model class and can be applied to *any* regression model.

- **Explicitly balancing calibration and sharpness.** While the pinball loss, as a proper scoring rule, targets both calibration and sharpness, the balance between these two quantities is made implicitly, which may result in a poor optimization objective. We propose a tunable loss function that targets calibration and sharpness separately, and allows the end-user to set an *explicit* balance.

- **Centered intervals.** In practice, we often desire uncertainty predictions made with centered intervals, which are not targeted via the pinball loss. We propose an alternative loss function that is better suited for this goal.

- **Encouraging individual calibration.** Perfect quantile forecasts will satisfy *individual* calibration (Eq. 2), which is a much stricter condition than the more-commonly used notion of *average* calibration (Eq. 3). We introduce a training procedure that aims to improve quantile predictions beyond average calibration, and demonstrate its efficacy via *adversarial group* calibration (Eq. 4).

We proceed by first describing methods of assessing the quality of predictive UQ and the pitfalls of optimizing the pinball loss in Section 2. Drawing motivation from this, we then present our proposed methods in Section 3. In Section 4, we demonstrate our methods experimentally, where we model predictive uncertainty on benchmark datasets, and on a high-dimensional, real-world uncertainty estimation task in the area of nuclear fusion.

## 2 Preliminaries and Background

We first lay out the notation, terminology, and class of models considered in this paper. Then we provide an overview of evaluation metrics in UQ and demonstrate how the pinball loss may be inadequate both as an evaluation metric and as an optimization objective.

### 2.1 Notation

Bold upper case letters $\mathbf{X}, \mathbf{Y}$ denote random variables, lower case letters $x, y$, denote their values, and calligraphic upper case letters $\mathcal{X}, \mathcal{Y}$ denote sets of possible values. We use $x \in \mathcal{X}$ to denote the input feature vector and $y \in \mathcal{Y}$ to denote the corresponding target. Additionally, we consider the regression setting where $\mathcal{Y} \subset \mathbb{R}$ and $\mathcal{X} \subset \mathbb{R}^n$. We use $\mathbb{F}_{\mathbf{X}}, \mathbb{F}_{\mathbf{Y}|x}, \mathbb{F}_{\mathbf{Y}}$ to denote the true cumulative distribution of the subscript random variable. For any $x \in \mathcal{X}$, we assume there exists a true conditional distribution $\mathbb{F}_{\mathbf{Y}|x}$ over $\mathcal{Y}$, and we assume $\mathbb{Q}_p(x)$ denotes the true $p^{\text{th}}$ quantile of this distribution, i.e. $\mathbb{F}_{\mathbf{Y}|x}(\mathbb{Q}_p(x)) = p$. Any estimates of the true functions $\mathbb{F}, \mathbb{Q}_p$ will be denoted with a hat, $\hat{\mathbb{F}}, \hat{\mathbb{Q}}_p$. We will specifically refer to any family of estimates for $\mathbb{Q}_p$, with $p \in (0, 1)$, as a "quantile model", denoted $\hat{\mathbb{Q}} : \mathcal{X} \times (0, 1) \to \mathcal{Y}$. Unless otherwise noted, we will always consider the *conditional* problem of estimating quantities in the target space $\mathcal{Y}$, conditioned on a value $x \in \mathcal{X}$.

### 2.2 Assessing the Quality of Predictive UQ

While various metrics have been proposed to assess the quality of UQ, there has been a great deal of recent focus on the notions of *calibration* and *sharpness* [15, 13, 65, 60, 55, 35, 21, 20]. We introduce calibration here, but for a more thorough treatment, see Zhao et al. [65]. Broadly speaking, calibration in the regression setting requires that the probability of observing the target random variable below a predicted $p^{\text{th}}$ quantile is equal to the *expected probability* $p$, for all $p \in (0, 1)$. We refer to the former quantity as the *observed probability* and denote it $p^{\text{obs}}(p)$, for an expected probability $p$, which we

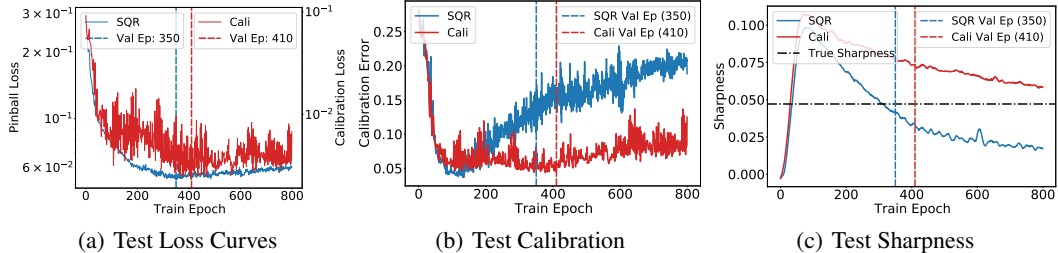

Figure 1: **(a)** Test loss continues to decrease until the validated epoch. **(b-c)** At the validated epoch, *SQR* (optimizes *pinball loss*) is highly miscalibrated while sharper than the true sharpness level. *Cali* (optimizes proposed *calibration loss*) is better calibrated while less sharp than the true sharpness.

will write as $p^{\text{obs}}$ when it is clear from context. Calibration requires $p^{\text{obs}}(p) = p, \forall p \in (0, 1)$. From this generic statement, we can describe different notions of calibration based on how $p^{\text{obs}}$ is defined.

A model is **individually calibrated** if it outputs the true conditional quantiles, i.e. $\hat{\mathbb{Q}}_p(x) = \mathbb{Q}_p(x)$. In this case, we define the observed probability to be

$$p^{\text{obs}}_{indv}(p, x) := \mathbb{F}_{\mathbf{Y}|x}(\hat{\mathbb{Q}}_p(x)), \quad \forall x \in \mathcal{X}, \quad \forall p \in (0, 1). \tag{2}$$

In words, this requires that the probability of observing $y$ below the quantile prediction is equal to $p$, *at each point $x \in \mathcal{X}$, individually*. If we can verify this property for all $x \in \mathcal{X}$, then by definition, we will know the quantile output is correct and precisely the true conditional quantile. However, individual calibration is typically unverifiable with finite datasets in the assumption-less case [65].

A relaxed condition is **average calibration**. In this case, we define the observed probability to be

$$p^{\text{obs}}_{avg}(p) := \mathbb{E}_{x \sim \mathbb{F}_{\mathbf{X}}}[\mathbb{F}_{\mathbf{Y}|x}(\hat{\mathbb{Q}}_p(x))], \quad \forall p \in (0, 1), \tag{3}$$

i.e. the probability of observing the target below the quantile prediction, *averaged over $\mathbb{F}_X$*, is equal to $p$. Average calibration is often referred to simply as "calibration" [13, 35]. Given a dataset $D = \{(x_i, y_i)\}_{i=1}^{N}$, we can estimate $p^{\text{obs}}_{avg}(p)$ with $\hat{p}^{\text{obs}}_{avg}(D, p) = \frac{1}{N} \sum_{i=1}^{N} \mathbb{I}\{y_i \leq \hat{\mathbb{Q}}_p(x_i)\}$. Note that if our quantile estimate achieves average calibration then $\hat{p}^{\text{obs}}_{avg} \to p$ as $N \to \infty, \forall p \in (0, 1)$. The degree of error in average calibration is commonly measured by *expected calibration error* [60, 13, 21], $\text{ECE}(D, \hat{\mathbb{Q}}) = \frac{1}{m} \sum_{j=1}^{m} |\hat{p}^{\text{obs}}_{avg}(D, p_j) - p_j|$, where $p_j \sim \text{Unif}(0, 1)$.

It may be possible to have an uninformative, yet average calibrated model. For example, quantile predictions that match the true *marginal* quantiles of $\mathbb{F}_{\mathbf{Y}}$ will be average calibrated, but will hardly be useful since they do not depend on the input $x$. Therefore, the notion of **sharpness** is also considered, which quantifies the concentration of distributional predictions [20]. For example, for non-parametric predictions, the width of a centered $95\%$ prediction interval is often used as a measure of sharpness. There generally exists a tradeoff between average calibration and sharpness [20, 46].

Recent works have suggested a notion of calibration stronger than average calibration, called adversarial group calibration [65]. This stems from the notion of **group calibration** [23, 29], which prescribes measurable subsets $\mathcal{S}_i \subset \mathcal{X}$ s.t. $P_{x \sim \mathbb{F}_{\mathbf{X}}}(x \in \mathcal{S}_i) > 0, i = 1, \ldots, k$, and requires the predictions to be average calibrated within each subset. Adversarial group calibration then requires average calibration for *any subset of $\mathcal{X}$ with non-zero measure*. Denote $\mathbf{X}_{\mathcal{S}}$ as a random variable that is conditioned on being in the set $\mathcal{S}$. For **adversarial group calibration**, the observed probability is

$$p^{\text{obs}}_{adv}(p) := \mathbb{E}_{x \sim \mathbb{F}_{\mathbf{X}_{\mathcal{S}}}}[\mathbb{F}_{\mathbf{Y}|x}(\hat{\mathbb{Q}}_p(x))], \quad \forall p \in (0, 1), \quad \forall \mathcal{S} \subset \mathcal{X} \text{ s.t. } P_{x \sim \mathbb{F}_{\mathbf{X}}}(x \in \mathcal{S}) > 0. \tag{4}$$

With a finite dataset, we can measure a proxy of adversarial group calibration by measuring the average calibration within all subsets of the dataset with sufficiently many points.

Intuitively, individual calibration inspects the discrepancy between $p^{\text{obs}}$ and $p$ for individual inputs $x \in \mathcal{X}$, adversarial group calibration relaxes this by inspecting any subset of $\mathcal{X}$ with non-zero measure, and average calibration relaxes this further by considering the full distribution of $\mathbf{X}$.

One alternative family of evaluation metrics is **proper scoring rules** [19]. Proper scoring rules are summary statistics of overall performance of a distributional prediction and consider both calibration

and sharpness jointly [20]. For example, negative log-likelihood (NLL) is a proper scoring rule that is commonly used with density predictions [14, 48, 37]. For quantile predictions, one proper score is the **check score**, *which is identical to the pinball loss*. Since proper scoring rules consider both calibration and sharpness together in a single value, they can serve as optimization objectives for UQ. For example, optimizing the pinball loss is the traditional method in quantile regression [31], and many recent quantile-based UQ methods focus on optimizing this objective [51, 58, 8, 63].

In this work, however, we note that the balance between calibration and sharpness implied by the pinball loss is arbitrary and depends on the expressivity of the model class—and with highly expressive models, this balance can be heavily skewed towards sharpness. In their seminal work on probabilistic forecasts, Gneiting and Raftery [19] contend that the goal of probabilistic forecasting is to "maximize the sharpness of the predictive distribution subject to calibration", i.e. calibration should be first achieved and then sharpness optimized. We show that common machine learning methods that use the pinball loss objective may in fact lead to an arbitrary and miscalibrated UQ.

**Proposition 1**. *Consider a finite dataset $D$, the pinball loss $\rho_\tau$ (Eq. 1) and a quantile model $f : \mathcal{X} \times (0,1) \to \mathcal{Y}$ that is average calibrated on $D$, i.e. $ECE(D, f) = 0$. Then there always exists another quantile model $g : \mathcal{X} \times (0,1) \to \mathcal{Y}$, such that, for any quantile level $\tau \in (0,1)$, $g$ has lower pinball loss than $f$ on $D$, i.e. $\sum_{i=1}^{N} \rho_\tau(y_i, g_\tau(x_i)) < \sum_{i=1}^{N} \rho_\tau(y_i, f_\tau(x_i))$, but worse average calibration than $f$, i.e. $ECE(D, g) > ECE(D, f)$.*

**Proof:** The proof is given in Appendix A.1.

This proposition essentially states how the pinball loss can become detached from calibration, and we show its practical ramifications via a synthetic example in Figure 1 (experiment details in Appendix E.1). We first note in Figure 1 (a) and (b) that even while the pinball loss decreases on the test set, test calibration worsens (while sharpness improves). Further, at the best validation epoch, optimizing the pinball loss converges to a solution that is sharper than the true noise level. Note that a UQ that is sharper than the true noise level will *never* be calibrated (meanwhile, a less sharp prediction *can still be calibrated*, e.g. the marginal distribution $\mathbb{F}_{\mathbf{Y}}$). While this may seem like an issue that can simply be addressed with regularization, we demonstrate in Appendix E.2 how that is not the case. These pitfalls motivate our methods in Section 3.

## 3 Methods

We propose four methods that aim to produce an improved quantile model. The first is a model-agnostic procedure that relies on conditional density estimation (Section 3.1). To address settings where density estimation may be difficult, we then propose two loss functions to optimize with differentiable models: the combined calibration loss (Section 3.2), which directly optimizes calibration and sharpness, and the interval score (Section 3.3), which is a proper scoring rule for centered intervals. Finally, we propose a group batching method (Section 3.4) that can be applied to the batch optimization procedure for any loss function (e.g. combined calibration loss, interval score, and even pinball loss) to induce better convergence towards adversarial group calibration.

### 3.1 Utilizing Conditional Density Estimation for Model Agnostic QR

One drawback of many existing quantile-based UQ methods is that their training procedure requires differentiable models. In fact, most UQ methods require a specific class of models because of their modeling structure or their loss objective (e.g. Gaussian processes [50], dropout [17], latent variable models [33], simultaneous pinball loss [58], and NLL-based losses [37]). This model restriction can be especially unfavorable in practical settings. A domain expert with an established point prediction model and compute infrastructure may want to add UQ without much additional overhead.

To address these issues, we can consider the following model-agnostic procedure. Instead of optimizing a designated loss function, we can consider splitting the given problem into two parts: estimate conditional quantiles directly from data, then regress onto these estimates. The benefit of this method is that, granted we can estimate the conditional quantiles accurately, we can use any regression model to regress onto these quantile estimates. Further, this regression task directly targets the goal of producing the true conditional quantiles (i.e. individual calibration). This procedure, which we refer to as *Model Agnostic QR* (MAQR), is outlined in Algorithm 1.

**Algorithm 1** MAQR

1: **Input:** Train data $\{x_i, y_i\}_{i=1}^N$, trained regression model $\hat{f}(x)$
2: Calculate residuals $\epsilon_i = y_i - \hat{f}(x_i)$, $i \in [N]$, and denote the residual dataset $R = \{x_i, \epsilon_i\}_{i=1}^N$
3: Initialize $D \leftarrow \varnothing$
4: **for** $i = 1$ **to** $N$ **do**
5: $\quad D_i \leftarrow$ CONDQUANTILESESTIMATORS$(R, i)$ (Algorithm 2)
6: $\quad D \leftarrow D \cup D_i$
7: **end for**
8: Use $D$ to fit a regression model $\hat{g}$
$\quad\quad \hat{g} : (x, p) \mapsto \epsilon$
9: **Output:** $\hat{f} + \hat{g}$

**Algorithm 2** CONDQUANTILESESTIMATORS

1: **Input:** Dataset $\{x_i, \epsilon_i\}_{i=1}^N$, point index $k \in [N]$
2: $E_{k,d_N} \leftarrow \{\epsilon_i \; : \; \text{dist}(x_k, x_i) \leq d_N, i \in [N]\}$
3: Construct an empirical CDF with $E_{k,d_N}$ to produce $\hat{\mathbb{F}}_{\mathbf{E}|x_k} : \epsilon \mapsto p \in [0, 1]$
4: Initialize $D \leftarrow \varnothing$
5: **for** each $\epsilon_j$ in $E_{k,d_N}$ **do**
6: $\quad \hat{p}_{k,j} \leftarrow \hat{\mathbb{F}}_{\mathbf{E}|x_k}(\epsilon_j)$
7: $\quad D \leftarrow D \cup \{x_k, \hat{p}_{k,j}, \epsilon_j\}$
8: **end for**
9: **Output:** $D$

MAQR is based on the key assumption that nearby points in $\mathcal{X}$ will have similar conditional distributions, i.e. if $x_j \approx x_k$ then $\mathbb{F}_{\mathbf{Y}|x_j} \approx \mathbb{F}_{\mathbf{Y}|x_k}$. Given this smoothness assumption, we can group neighboring points to estimate the conditional density at each locality over $\mathcal{X}$, with locality determined by the hyperparameter $d_N$ (Algorithm 2, line 2). We then construct an empirical CDF with the group of neighboring points, and conditional quantile estimates are produced with this empirical CDF. These estimates are collected into $D$ (Algorithm 1, line 6), which is ultimately used as the training set for the quantile model $\hat{g}$.

In practice, we perform these steps with *residuals*, by first estimating a mean function $\hat{f}$ (Algorithm 1, line 1). This practical choice stems from existing works in conditional density estimation, which suggests that having 0 conditional mean in the data provides benefits in terms of lower asymptotic mean squared error in the conditional density predictions [26]. Further, this demonstrates how MAQR can be readily applied in the application setting where an accurate point prediction model often already exist.

Algorithm 1 is a specific implementation of a more general model-agnostic algorithm, in which we directly estimate conditional quantiles from the data with tools from conditional density estimation. We note that using KDEs for conditional density estimation is a well studied problem with theoretical guarantees [25, 26, 57]. In the case the distance in $\mathcal{X}$ is measured using a uniform kernel with mild assumptions on the bandwidth, Algorithm 1 falls under the guarantees stated by Stute et al. [57].

**Theorem 1** [57]. *Assume $\mathcal{Y} \subset \mathbb{R}$, $\mathcal{X} \subset \mathbb{R}^n$, $\text{dist}(x_i, x_j) := |x_i - x_j|_\infty$, and that $\hat{\mathbb{F}}_{\mathbf{E}|x}$ is constructed using the procedure given in line 5 of Algorithm 1 (i.e. $x_i = x$). Further assume that, as $N \to \infty$, $d_N \to 0$ and that $\sum_{N \geq 1} \exp(-\rho N d_N^n) < \infty$, $\forall \rho > 0$. Then, as $N \to \infty$, for almost all $x \in \mathcal{X}$, $\sup_\epsilon [\hat{\mathbb{F}}_{\mathbf{E}|x}(\epsilon) - \mathbb{F}_{\mathbf{E}|x}(\epsilon)] \to 0$ with probability 1.*

This theorem states that in the limit of data, for almost all $x \in \mathcal{X}$, the CDF estimate $\hat{\mathbb{F}}_{\mathbf{E}|x}$ will converge uniformly to the true CDF $\mathbb{F}_{\mathbf{E}|x}$ with probability 1. The dataset, $D$, will therefore be populated with good estimates of the conditional quantile and quantile level pair for $x$. In Appendix B, we state the general form of Algorithm 1 and also demonstrate how the algorithm is model agnostic. Through our experiments in Section 4, we will show empirically that utilizing these density estimates sidesteps the issues inherent to the pinball loss and produces much higher quality quantile predictions.

### 3.2 Explicitly balancing calibration and sharpness with the combined calibration loss

While MAQR can produce strong results, its performance can suffer in high-dimensional settings, where nonparametric conditional density estimation methods falter. Neural networks (NNs) have shown good performance in high dimensional settings, given their high capacity to approximate complex functions and recent advances in fast gradient-based optimization. We therefore propose a loss-based approach to estimating conditional quantiles for NNs and other differentiable models.

Drawing motivation from the *arbitrary* balance between calibration and sharpness that pinball loss *implicitly* provides, we propose objectives separately for calibration and sharpness, Then, we combine the two objectives into a single loss function that provides an explicit balance between calibration and sharpness that can be chosen by the end user.

We first consider calibration of a quantile prediction, $\hat{\mathbb{Q}}_p \in \mathcal{Y}$ for quantile level $p \in (0, 1)$. Here, we omit conditioning on $x$ for clarity. For this prediction to be average calibrated, exactly a $p$ proportion of the true density should lie below $\hat{\mathbb{Q}}_p$, i.e. $p_{avg}^{\text{obs}} = P(Y \leq \hat{\mathbb{Q}}_p) = p$. While calibration (e.g. $|p_{avg}^{\text{obs}} - p|$) is a non-differentiable objective, by inducing a truncated distribution based on the current level of calibration, we can construct the following calibration objective, which is minimized if and only if the prediction is average calibrated:

$$\mathcal{C}(\hat{\mathbb{Q}}_p, p) = \mathbb{I}\{\hat{p}_p < p\} * \mathbb{E}[Y - \hat{\mathbb{Q}}_p | Y > \hat{\mathbb{Q}}_p] * P(Y > \hat{\mathbb{Q}}_p) \tag{5}$$
$$+ \mathbb{I}\{\hat{p}_p > p\} * \mathbb{E}[\hat{\mathbb{Q}}_p - Y | \hat{\mathbb{Q}}_p > Y] * P(\hat{\mathbb{Q}}_p > Y), \text{where } \hat{p}_p = P(Y \leq \hat{\mathbb{Q}}_p).$$

The empirical calibration objective, $\mathcal{C}(D, \hat{\mathbb{Q}}_p, p)$, is then defined as follows:

$$\mathcal{C}(D, \hat{\mathbb{Q}}, p) = \mathbb{I}\{\hat{p}_{avg}^{\text{obs}} < p\} * \frac{1}{N} \sum_{i=1}^{N} \left[ (y_i - \hat{\mathbb{Q}}_p(x_i))\mathbb{I}\{y_i > \hat{\mathbb{Q}}_p(x_i)\} \right]$$
$$+ \mathbb{I}\{\hat{p}_{avg}^{\text{obs}} > p\} * \frac{1}{N} \sum_{i=1}^{N} \left[ (\hat{\mathbb{Q}}_p(x_i) - y_i)\mathbb{I}\{\hat{\mathbb{Q}}_p(x_i) > y_i\} \right]. \tag{6}$$

***Note 1: Intuition of the calibration objective.*** For any given $p$, consider the case when the quantile estimate $\hat{\mathbb{Q}}_p$ is below the true $p^{\text{th}}$ quantile $\mathbb{Q}_p$. Since $\hat{\mathbb{Q}}_p < \mathbb{Q}_p \implies \hat{p}_p < p$, this implies that too much data density lies above $\hat{\mathbb{Q}}_p$. In this case, $\mathcal{C}(\hat{\mathbb{Q}}_p, p)$ reduces to $\mathbb{E}[Y - \hat{\mathbb{Q}}_p | Y > \hat{\mathbb{Q}}_p] * P(Y > \hat{\mathbb{Q}}_p)$. $\hat{\mathbb{Q}}_p$ is pulled higher with the expectation of the truncated distribution that places $\hat{\mathbb{Q}}_p$ at the lower bound of the support. In the opposite case, when $\hat{\mathbb{Q}}_p > \mathbb{Q}_p$, $\hat{\mathbb{Q}}_p$ is pulled lower by the same logic.

***Note 2: Is the proposed calibration objective a proper scoring rule?*** Strictly speaking, the calibration objective is a non-decomposable function, hence deviates from the standard convention of proper scoring rules [19], which can be "decomposed" into scores for individual examples $(x_i, y_i)$. This simply arises from the fact that measuring average calibration (i.e. $\hat{p}_{avg}^{\text{obs}}$) is non-decomposable. Proper scoring rules are defined such that an optimum of the *expected score* (or *risk*, if we consider the score as a *loss function*) occurs at the true distribution quantity. While an example level *loss* or *score* does not exist due to non-decomposability, we can still show the (expectation-level) score (i.e. $\mathcal{C}(\hat{\mathbb{Q}}_p, p)$) is minimized by the true distribution and hence enjoys the optimum property of proper scoring rules.

**Proposition 2**. *For any quantile level $p \in (0, 1)$, the true quantile function $\mathbb{Q}_p$ minimizes the calibration objective, $\mathcal{C}(\hat{\mathbb{Q}}_p, p)$. Further, on a finite dataset $D$, the empirical calibration objective, $\mathcal{C}(D, \hat{\mathbb{Q}}_p, p)$, is minimized by an average calibrated solution on $D$, i.e. when $\hat{p}_{avg}^{obs}(D, p) = p$.*

**Proof:** The proof is given in Appendix A.2.

***Note 3: Non-zero gradients for miscalibrated predictions $\hat{\mathbb{Q}}_p$.*** We can further show that for a miscalibrated quantile prediction, the gradients of $\mathcal{C}$ are always non-zero. When $\hat{p}_p < p$, $\partial\mathcal{C}(\hat{\mathbb{Q}}_p, p)/\partial\hat{\mathbb{Q}}_p = -P(Y > \hat{\mathbb{Q}}_p) < 0$. Thus increasing $\hat{\mathbb{Q}}_p$ decreases the objective $\mathcal{C}$. Similarly, when $\hat{p}_p > p$, $\partial\mathcal{C}(\hat{\mathbb{Q}}_p, p)/\partial\hat{\mathbb{Q}}_p = P(Y < \hat{\mathbb{Q}}_p) > 0$, and an analogous argument follows (proof in Appendix A.3).

As discussed in Section 2.2, average calibration by itself is not a sufficient condition for meaningful UQ, hence we also desire *sharp* quantile models, with more-concentrated (less dispersed) distributions. We can induce this property in quantile predictions by predicting the $(1 - p)^{\text{th}}$ quantile $\hat{\mathbb{Q}}_{1-p}(x_i)$ alongside each prediction $\hat{\mathbb{Q}}_p(x_i)$ and penalizing the width between the quantile predictions:

$$\mathcal{P}(\hat{\mathbb{Q}}_p, p) = \mathbb{E}\left[\left|\hat{\mathbb{Q}}_p - \hat{\mathbb{Q}}_{1-p}\right|\right]. \tag{7}$$

The empirical sharpness objective, $\mathcal{P}(D, \hat{\mathbb{Q}}_p, p)$, is then defined as follows:

$$\mathcal{P}(D, \hat{\mathbb{Q}}, p) = \frac{1}{N} \sum_{i=1}^{N} \begin{cases} \hat{\mathbb{Q}}_{1-p}(x_i) - \hat{\mathbb{Q}}_p(x_i) & (p \leq 0.5) \\ \hat{\mathbb{Q}}_p(x_i) - \hat{\mathbb{Q}}_{1-p}(x_i) & (p > 0.5). \end{cases} \tag{8}$$

It is important to note that the true underlying distribution will not have $0$ sharpness if there is significant noise, and sharpness should be optimized subject to calibration. Therefore, we should only penalize sharpness when the data suggests our quantiles are too dispersed, i.e. when $\left| p_{avg}^{\text{obs}}(p) - p_{avg}^{\text{obs}}(1-p) \right|$, the *observed coverage* between the pair of quantiles $\hat{\mathbb{Q}}_p(x_i)$ and $\hat{\mathbb{Q}}_{1-p}(x_i)$, is greater than $|2p-1|$, the *expected coverage*.

Combining the calibration and sharpness terms, we have the **combined calibration loss**

$$\mathcal{L}(D, \hat{\mathbb{Q}}_p, p) = (1-\lambda)\mathcal{C}(D, \hat{\mathbb{Q}}_p, p) + \lambda\mathcal{P}(D, \hat{\mathbb{Q}}_p, p). \tag{9}$$

The hyperparameter $\lambda \in [0,1]$ sets the explicit balance between calibration and sharpness. Note that setting $\lambda = 0$ may not always be desirable, since optimizing $\mathcal{C}(D, \hat{\mathbb{Q}}_p, p)$ alone may converge to quantiles of the marginal distribution, $\mathbb{F}_{\mathbf{Y}}$. Further, in certain downstream applications that utilize UQ, a sharper prediction, even at the cost of worse calibration, may result in higher utility, and $\lambda$ can be tuned according to the utility function of the application. In our experiments, we tune $\lambda$ by cross-validating with adversarial group calibration as it is the strictest notion of calibration that can be estimated with a finite dataset. Since we learn a quantile model that outputs the conditional quantile estimates for all probabilities, our training objective is $\mathbb{E}_{p \sim \text{Unif}(0,1)}\mathcal{L}(D, \hat{\mathbb{Q}}_p, p)$.

### 3.3 Encouraging calibration of centered intervals with the interval score

The combined calibration loss (Eq. 9) optimizes average calibration, which targets observed probabilities below a quantile. In many applications, however, we may desire a centered prediction interval (PI) which requires a pair of quantile predictions. A centered $95\%$ PI, for example, is a pair of quantile predictions at quantile levels $0.025$ and $0.975$. Hence, for the average calibration of the $p^{\text{th}}$ *centered interval*, we want $\left[ \hat{p}_{avg}^{\text{obs}}(0.5 + \frac{p}{2}) - \hat{p}_{avg}^{\text{obs}}(0.5 - \frac{p}{2}) \right]$ (the PI's observed probability, a.k.a. prediction interval coverage probability (PICP) [58, 28, 49]) to be equal to the expected probability $p$. While we can modify the objective in Eq. 9 to adhere to this altered goal, here we propose simultaneously optimizing the **interval score** (or Winkler score) [19, 62] for all expected probabilities $p \in (0,1)$, and bring to light a proper scoring rule that has largely been neglected for the purpose of *learning quantiles*. While some previous works utilize the interval score to *evaluate* interval predictions [7, 4, 1, 40], to the best of our knowledge, no previous work has focused on simultaneously optimizing it and shown a thorough experimental evaluation as we provide in Section 4.

For a point $(x, y)$, if we denote a $(1-\alpha)$ centered PI as $\hat{l}, \hat{u}$, i.e. $\hat{l} = \hat{\mathbb{Q}}_{\frac{\alpha}{2}}(x)$ and $\hat{u} = \hat{\mathbb{Q}}_{1-\frac{\alpha}{2}}(x)$, the interval score is defined as $S_\alpha(\hat{l}, \hat{u}; y) = (\hat{u} - \hat{l}) + \frac{2}{\alpha}(\hat{l} - y)\mathbb{I}\{y < \hat{l}\} + \frac{2}{\alpha}(y - \hat{u})\mathbb{I}\{y > \hat{u}\}$. We show in Appendix A.4 that the minimum of the expectation of the interval score is attained at the true conditional quantiles, $\hat{l} = \mathbb{Q}_{\frac{\alpha}{2}}(\cdot)$, $\hat{u} = \mathbb{Q}_{1-\frac{\alpha}{2}}(\cdot)$. We train our quantile model for all centered intervals (and hence all quantile levels) simultaneously by setting our loss as $\mathbb{E}_{\alpha \sim \text{Unif}(0,1)}S_\alpha$.

### 3.4 Inducing adversarial group calibration with group batching

The calibration loss (Section 3.2) and the interval score (Section 3.3) optimize for the *average* calibration of quantiles and centered intervals, respectively. To get closer to individual calibration, one condition we can additionally require is *adversarial group calibration*. Since adversarial group calibration requires average calibration over any subset of non-zero measure over the domain, this is not fully observable with finite datasets $D$ for all subset sizes. However, for any subset in $D$ with enough datapoints, we can still estimate average calibration over the subset. Hence, we can apply our optimization objectives onto appropriately large subsets to induce adversarial group calibration.

In practice, this involves constructing subsets within the domain and taking gradient steps based on the loss over each subset. In naive implementations of stochastic gradient descent, a random batch is drawn *uniformly* from the training dataset $D$, and a gradient step is taken according to the loss over this batch. This is also the case in *SQR* [58]. The uniform draw of the batch will tend to preserve $\mathbb{F}_{\mathbf{X}}$ (the marginal distribution of $\mathbf{X}$), hence optimizing average calibration over this batch will only induce average calibration of the model.

Instead, deliberately grouping the datapoints based on input features, and then batching and taking gradient steps based on these batches, induces better adversarial group calibration. We find in our experiments that adversarial group calibration improves significantly with simple implementations of group batching, and in Section 4.2, we show through an ablation study that group batching can improve average calibration and adversarial group calibration of *SQR* as well.

| | | SQR | mPAIC | *Interval* | *Cali* | *MAQR* |
|---|---|---|---|---|---|---|
| **UCI** | Concrete | $9.3 \pm 1.5(\underline{7.0} \pm 1.0)$ | $6.2 \pm 0.5(14.2 \pm 0.8)$ | $\mathbf{3.7} \pm 0.6(18.1 \pm 0.6)$ | $5.6 \pm 0.8(17.3 \pm 1.5)$ | $5.3 \pm 0.4(16.0 \pm 0.4)$ |
| | Power | $2.6 \pm 0.4(13.4 \pm 0.2)$ | $5.2 \pm 0.4(13.5 \pm 0.3)$ | $2.2 \pm 0.4(21.0 \pm 1.0)$ | $2.0 \pm 0.1(\underline{13.1} \pm 0.1)$ | $\mathbf{1.6} \pm 0.3(19.9 \pm 0.2)$ |
| | Wine | $4.2 \pm 0.2(29.5 \pm 0.4)$ | $10.3 \pm 0.3(37.7 \pm 0.5)$ | $5.0 \pm 0.8(41.4 \pm 2.5)$ | $4.2 \pm 0.4(\underline{26.0} \pm 0.8)$ | $\mathbf{2.7} \pm 0.2(39.3 \pm 0.5)$ |
| | Yacht | $9.4 \pm 0.9(\underline{1.0} \pm 0.1)$ | $10.8 \pm 2.3(2.6 \pm 0.4)$ | $7.5 \pm 0.9(4.5 \pm 1.0)$ | $8.3 \pm 0.6(2.0 \pm 0.4)$ | $\mathbf{6.8} \pm 2.1(2.4 \pm 0.3)$ |
| | Naval | $9.7 \pm 1.6(3.5 \pm 0.4)$ | $3.1 \pm 0.5(63.0 \pm 1.8)$ | $4.7 \pm 1.4(28.4 \pm 3.6)$ | $5.9 \pm 0.7(3.0 \pm 0.2)$ | $\mathbf{2.3} \pm 0.2(\underline{1.7} \pm 0.1)$ |
| | Energy | $9.8 \pm 0.8(\underline{2.0} \pm 0.1)$ | $10.4 \pm 0.5(4.3 \pm 0.2)$ | $4.3 \pm 0.6(5.1 \pm 0.9)$ | $5.8 \pm 0.4(3.6 \pm 0.3)$ | $\mathbf{3.5} \pm 1.0(3.2 \pm 0.1)$ |
| | Boston | $9.0 \pm 0.8(\underline{9.3} \pm 0.7)$ | $8.7 \pm 1.3(12.3 \pm 0.7)$ | $6.9 \pm 1.1(20.3 \pm 0.5)$ | $8.5 \pm 1.5(10.9 \pm 0.6)$ | $\mathbf{6.2} \pm 1.8(10.9 \pm 0.8)$ |
| | Kin8nm | $4.4 \pm 0.1(\underline{11.4} \pm 0.2)$ | $6.6 \pm 0.4(17.0 \pm 0.5)$ | $2.9 \pm 0.4(16.9 \pm 0.5)$ | $3.5 \pm 0.3(13.7 \pm 0.7)$ | $\mathbf{1.8} \pm 0.4(17.1 \pm 0.1)$ |

Figure 2: **UCI Experiments. (Top Table):** Average calibration (measured by ECE) and sharpness in parentheses. The best mean ECE for each dataset has been bolded and the best mean sharpness has been underlined (all values multiplied by 100 for readability). **(Bottom Figure):** Adversarial group calibration for the first 5 UCI datasets (full set of results in Appendix D.1). Group size refers to proportion of test dataset size.

| | | SQR | mPAIC | *Interval* | *Cali* |
|---|---|---|---|---|---|
| **Fusion** | aminor | $5.8 \pm 0.9(\underline{1.6} \pm 0.0)$ | $13.5 \pm 0.0(5.3 \pm 0.0)$ | $3.6 \pm 0.7(3.7 \pm 0.1)$ | $\mathbf{2.9} \pm 0.2(2.2 \pm 0.0)$ |
| | betan | $\mathbf{3.1} \pm 0.4(\underline{2.8} \pm 0.1)$ | $9.2 \pm 0.4(5.5 \pm 0.1)$ | $5.1 \pm 0.4(5.7 \pm 0.3)$ | $3.4 \pm 0.5(3.3 \pm 0.2)$ |
| | dssdenest | $4.4 \pm 0.5(\underline{7.2} \pm 0.2)$ | $8.4 \pm 0.1(13.7 \pm 0.1)$ | $\mathbf{2.9} \pm 0.3(13.3 \pm 0.4)$ | $4.1 \pm 0.5(8.9 \pm 0.3)$ |
| | ip | $3.2 \pm 0.3(\underline{2.4} \pm 0.1)$ | $13.5 \pm 0.2(9.0 \pm 0.2)$ | $4.4 \pm 0.3(5.4 \pm 0.1)$ | $\mathbf{2.3} \pm 0.2(3.8 \pm 0.1)$ |
| | kappa | $4.4 \pm 0.6(\underline{2.5} \pm 0.1)$ | $14.8 \pm 0.5(9.0 \pm 0.4)$ | $4.2 \pm 0.4(6.1 \pm 0.2)$ | $\mathbf{3.6} \pm 0.2(3.4 \pm 0.1)$ |
| | li | $4.4 \pm 0.6(\underline{1.0} \pm 0.1)$ | $12.6 \pm 0.6(2.7 \pm 0.1)$ | $3.7 \pm 0.6(2.2 \pm 0.1)$ | $\mathbf{2.9} \pm 0.4(1.3 \pm 0.1)$ |
| | R0 | $5.3 \pm 0.8(\underline{3.3} \pm 0.1)$ | $8.7 \pm 0.4(7.2 \pm 0.3)$ | $\mathbf{3.4} \pm 0.2(6.2 \pm 0.1)$ | $3.6 \pm 0.2(4.0 \pm 0.3)$ |
| | tribot | $4.6 \pm 0.4(\underline{2.4} \pm 0.1)$ | $15.1 \pm 0.7(8.8 \pm 0.7)$ | $\mathbf{3.9} \pm 0.5(6.0 \pm 0.2)$ | $4.6 \pm 0.5(3.0 \pm 0.2)$ |
| | tritop | $5.6 \pm 0.7(\underline{2.7} \pm 0.1)$ | $12.9 \pm 0.7(7.2 \pm 0.7)$ | $3.7 \pm 0.8(6.5 \pm 0.4)$ | $\mathbf{2.5} \pm 0.3(4.4 \pm 0.1)$ |
| | volume | $5.8 \pm 1.2(\underline{0.9} \pm 0.0)$ | $16.9 \pm 1.1(3.9 \pm 0.4)$ | $3.6 \pm 0.3(2.0 \pm 0.1)$ | $\mathbf{2.8} \pm 0.1(1.2 \pm 0.0)$ |

Figure 3: **Fusion Experiments. (Top Table):** Average calibration (measured by ECE) and sharpness in parentheses. The best mean ECE for each dataset has been bolded and the best mean sharpness has been underlined (all values multiplied by 100 for readability). **(Bottom Figure):** Adversarial group calibration for the first 5 fusion datasets (full set of results in Appendix D.2). Group size refers to proportion of test dataset size.

To summarize, the main idea we introduce here with group batching is that, only taking uniform batches from the training set (thus only drawing batches which preserve $\mathbb{F}_\mathbf{X}$) can be detrimental when optimizing for calibration. Thus, additionally drawing batches based on deliberate groupings within the training set (thus, batches which do not preserve $\mathbb{F}_\mathbf{X}$) can help to induce a stronger notion of calibration (i.e. adversarial group calibration) in the model than average calibration. This concept is quite general and allows for variations in implementations when constructing the groups. In Appendix C.3 we provide details on how we implemented group batching for our experiments and ablation study.

## 4   Experiments

We demonstrate the performances of our proposed methods on the standard 8 UCI datasets [2], and on a real-world problem in nuclear fusion. To assess the predictions, we use all metrics from Section 2.2 that can be estimated with a finite test dataset: 1) average calibration vs sharpness, 2) adversarial group calibration, 3) centered interval calibration, 4) check score, and 5) interval score. The results for the first two of these metrics are displayed in the main body of the paper, and the results for the

| | Cali | | SQR | |
|---|---|---|---|---|
| | *Random Batch* | **Group Batch** | *Random Batch* | **Group Batch** |
| Boston | $9.7 \pm 1.3(\underline{10.2} \pm 0.7)$ | $\mathbf{8.5} \pm 1.5(10.9 \pm 0.6)$ | $10.9 \pm 1.0(\underline{8.8} \pm 1.1)$ | $\mathbf{9.8} \pm 1.2(9.5 \pm 0.9)$ |
| Concrete | $6.6 \pm 0.9(17.6 \pm 2.3)$ | $\mathbf{5.6} \pm 0.8(\underline{17.3} \pm 1.5)$ | $9.8 \pm 1.3(\underline{7.0} \pm 0.6)$ | $\mathbf{7.1} \pm 0.9(8.5 \pm 0.6)$ |
| Energy | $9.2 \pm 0.3(\underline{2.8} \pm 0.1)$ | $\mathbf{5.8} \pm 0.4(3.6 \pm 0.3)$ | $10.2 \pm 0.8(\underline{1.8} \pm 0.1)$ | $\mathbf{6.9} \pm 1.1(2.4 \pm 0.2)$ |
| Kin8nm | $\mathbf{3.4} \pm 0.3(\underline{13.7} \pm 0.4)$ | $3.5 \pm 0.3(\underline{13.7} \pm 0.7)$ | $4.7 \pm 0.3(\underline{11.1} \pm 0.1)$ | $\mathbf{3.9} \pm 0.4(11.3 \pm 0.2)$ |
| Wine | $4.4 \pm 0.5(\underline{25.6} \pm 0.8)$ | $\mathbf{4.2} \pm 0.4(26.0 \pm 0.8)$ | $4.5 \pm 0.4(29.7 \pm 0.5)$ | $\mathbf{4.0} \pm 0.4(\underline{28.5} \pm 0.8)$ |
| Yacht | $11.1 \pm 1.8(\underline{1.8} \pm 0.1)$ | $\mathbf{8.3} \pm 0.6(2.0 \pm 0.4)$ | $9.0 \pm 0.9(\underline{0.9} \pm 0.1)$ | $\mathbf{8.9} \pm 0.9(2.3 \pm 0.2)$ |
| Power | $\mathbf{1.7} \pm 0.2(14.2 \pm 0.3)$ | $2.0 \pm 0.1(\underline{13.1} \pm 0.1)$ | $\mathbf{2.5} \pm 0.3(14.0 \pm 0.5)$ | $2.9 \pm 0.5(\underline{13.6} \pm 0.8)$ |
| Naval | $2.8 \pm 0.2(\underline{12.1} \pm 3.1)$ | $\mathbf{2.4} \pm 0.3(50.6 \pm 8.6)$ | $8.6 \pm 1.6(\underline{3.6} \pm 0.1)$ | $\mathbf{5.3} \pm 0.8(6.0 \pm 0.5)$ |

Figure 4: **Group Batching Ablation: Average Calibration and Sharpness.** The table shows mean ECE and sharpness (in parentheses) and their standard error with and without group batching. The best mean ECE for each dataset has been bolded and the best mean sharpness has been underlined for *Cali* and *SQR* separately. All values have been multiplied by 100 for readability.

other three metrics are shown in Appendix D due to space restrictions. We describe how each of these evaluation metrics are calculated in Appendix C.4.

We provide comparisons against current state-of-the-art UQ methods for which computing the above metrics is tractable. *SQR* [58] is an NN model that optimizes the pinball loss for a batch of random quantile levels $p \sim \mathrm{Unif}(0, 1)$. *mPAIC* [65] is an extension of probabilistic neural networks [37, 47] that optimizes a combination of the standard Gaussian NLL and a loss that induces individual calibration. While additional quantile and PI based UQ methods exist, they are mostly designed to output a single quantile level or interval coverage level, which makes measuring calibration extremely expensive (training up to 100 models separately). For these methods, we provide a comparison on a simplified task of predicting the $95\%$ PI in Appendix E.3. Lastly, we could also consider the recalibration algorithm of Kuleshov et al. [35], which is not a standalone UQ method, but a post-hoc refinement step that can be applied on top of other methods. We discuss recalibration results in Appendix E.4. All results report the mean and error across 5 trials. Error bars and shaded bands in plots indicate $\pm 1$ standard error.

## 4.1 UCI and Fusion Experiments

**UCI datasets:** We evaluate 5 methods on the 8 UCI benchmark datasets: three proposed algorithms—*MAQR*, *Cali* (combined calibration loss), and *Interval* (interval score)—and 2 alternative algorithms—*SQR* and *mPAIC*. Appendix C.1 includes more details on the experiment setup and hyperparameters.

**Fusion datasets:** We further evaluate the methods on a high-dimensional task from nuclear fusion: quantifying uncertainty in plasma dynamics. Recently, there has been increasing interest in applying machine learning to prediction and control tasks in the area of nuclear fusion [11, 9, 43]. The plasma data in this paper was recorded from the DIII-D tokamak, a nuclear fusion device operated by General Atomics [39]. Plasma dynamics during fusion reactions are highly stochastic and running live fusion experiments is costly. Hence, practitioners use this dataset to learn a dynamics model of the system for various purposes, such as controller learning to optimize reaction efficiency and stability [16, 5, 6]. There are 10 scalar target signals to model in this dataset (full list in Appendix C.2), each describing a particular aspect of the current state of plasma. For each signal, the input features are 468 dimensional. We do not apply *MAQR* on this dataset because of the high computational costs and statistical challenge associated with nonparametric density estimation in high dimensions. Appendix C.2 provides more details on the dataset, experiment setup, and hyperparameters.

**Analysis of Results:** Figure 2 displays average calibration-sharpness for all 8 UCI datasets and adversarial group calibration for the first 5 UCI datasets (alphabetical order). *MAQR* produces the best average calibrated models on 7 of the 8 UCI datasets, and adversarial group calibration also indicates that *MAQR* tends to achieve the lowest calibration error across *any* random subgroup of *any* size with more than one point (6 out of 8 datasets). Excluding *MAQR*, *Interval* and *Cali* achieve competitive average calibration and adversarial group calibration on 7 out of 8 UCI datasets. Notably, *SQR* tends to produce the sharpest predictions across all datasets (often at the cost of worse calibration), which is not surprising following the discussion in Section 2.2. Lastly, we also note that *Interval* and *Cali* tend to be less brittle compared to *SQR* and *mPAIC*, which incur major failures in some cases (e.g. *mPAIC* in Kin8nm, *SQR* in Naval).

The fusion experiment results display a similar pattern (Figure 3). *Interval* and *Cali* achieve competitive average calibration and adversarial group calibration on 9 out of 10 fusion datasets, and *SQR* produces the sharpest UQ at the cost worse average and adversarial group calibration (except on betan). We also observe that *mPAIC* performs the poorest on the fusion datasets, with least calibrated and least sharp predictions. It is generally known that plasma dynamics display complex stochasticity [16, 34], and hence we suspect *mPAIC*'s performance degrades significantly because it assumes a Gaussian output and is trained according to the Gaussian likelihood.

Only a subset of the plots and metrics are shown in the main paper due to space restrictions. Readers are encouraged to see the full set of results in Appendix D, but we summarize the main findings here. The check score, interval score, and centered interval calibration in the tables in Appendix D.1 also rank *MAQR*'s prediction as best on the UCI datasets. This is surprising since *SQR* and *Interval* train NNs of the same capacity to *explicitly minimize* the check and interval scores, respectively. This indicates the distribution predicted by *MAQR* is fundamentally different as it utilizes direct estimates of the conditional distribution, while the other methods all optimize a specific loss function. The centered interval calibration metric on the fusion datasets (Appendix D.2) indicates that *Cali* and *Interval* both produce competitive centered PI's (9 out of 10 fusion datasets), suggesting that both methods have generally estimated the conditional quantiles better than the alternative algorithms.

## 4.2 Ablation Study: Effect of Group Batching

We provide an ablation study on the effect of group batching (experiment details in Appendix F.1). Figure 4 displays how group batching affects average calibration-sharpness of *Cali* and *SQR* on all 8 UCI datasets. Average calibration improved on 6 out of 8 datasets for *Cali*, and on 7 out of 8 datasets on *SQR*. While sharpness tends to worsen with group batching, it's important to note that less sharp (i.e. more dispersed) predictions *are* desirable if there is high noise in the true data distribution. In Appendix F.2, we show that group batching also improves adversarial group calibration, suggesting that the underlying data distribution truly does have high noise. This study indicates that deliberately taking batches during training that do not follow $\mathbb{F}_{\mathbf{X}}$ can improve UQ performance.

## 5 Discussion

In this paper, we have proposed four methods to improve quantile estimates for calibrated uncertainty quantification in regression. We assert that the pinball loss may not be an adequate objective to optimize in order to achieve calibration, and that our proposed methods provide better means of learning calibrated conditional quantiles. We have also extended the scope of regression models on which quantile-based UQ can be applied by developing a model-agnostic method. This can be of practical interest to users that have specific training infrastructure or preexisting regression procedures, since these procedures can be leveraged to quantify uncertainty without much additional overhead.

By providing an extensive evaluation with a suite of metrics, we also aim to show that there is not one single metric that captures full information about the performance of a UQ procedure. Rather, a holistic review of multiple types of metrics sheds light on different aspects of performance. This point has motivated us to additionally develop and make publicly available *Uncertainty Toolbox* [12], a Python library for evaluation, visualization and recalibration of predictive uncertainty, which includes the full suite of metrics discussed in this work.

In certain applications, users may also be concerned with modeling epistemic uncertainty. This is somewhat orthogonal to the goals of this paper; nevertheless, we provide a discussion on how bootstrapped ensembles of our methods can incorporate epistemic uncertainty in Appendix G.

**Acknowledgments**

This work was funded in part by DOE grant number DE-SC0021414 and DE-AC02-76SF00515.

Willie Neiswanger was supported in part by NSF (#1651565), ONR (N000141912145), AFOSR (FA95501910024), ARO (W911NF-21-1-0125), DOE (DE-AC02-76SF00515) and Sloan Fellowship.

Ian Char was also supported by NSF grant DGE1745016. Any opinions, findings, and conclusions or recommendations expressed in this material are those of the author(s) and do not necessarily reflect the views of the National Science Foundation.

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
