# Appendix for "Beyond Pinball Loss: Quantile Methods for Calibrated Uncertainty Quantification"

## A  Theoretical Results

### A.1  Proof of Proposition 1

**Proposition 1**. *Consider a finite dataset $D$, the pinball loss $\rho_\tau$ (Eq. 1) and a quantile model $f : \mathcal{X} \times (0,1) \to \mathcal{Y}$ that is average calibrated on $D$, i.e. $ECE(D,f) = 0$. Then there always exists another quantile model $g : \mathcal{X} \times (0,1) \to \mathcal{Y}$, such that, for any quantile level $\tau \in (0,1)$, $g$ has lower pinball loss than $f$ on $D$, i.e. $\sum_{i=1}^{N} \rho_\tau(y_i, g_\tau(x_i)) < \sum_{i=1}^{N} \rho_\tau(y_i, f_\tau(x_i))$, but worse average calibration than $f$, i.e. $ECE(D,g) > ECE(D,f)$.*

**Proof:**

Denote $\{p_j\}_{j=1}^{m}$ as the set of quantiles used to compute $ECE(D,f)$. Since $f$ is average calibrated on $D$, for any $\tau \in \{p_j\}_{j=1}^{m}$, $\hat{p}_{avg}^{obs}(D,\tau)$ of $f$ is equal to $\tau$, i.e. $\frac{1}{N}\sum_{i=1}^{N} \mathbb{I}\{y_i \le f_\tau(x_i)\} = \tau$.

Hence,

- $|\{(x_i,y_i) \mid y_i \le f_\tau(x_i)\}| = \tau * N$,

- $|\{(x_i,y_i) \mid y_i > f_\tau(x_i)\}| = (1-\tau) * N$.

Denote $\{(x_i,y_i) \mid y_i \le f_\tau(x_i)\}$ as $S_f^{\text{under}}$ and $\{(x_i,y_i) \mid y_i > f_\tau(x_i)\}$ as $S_f^{\text{over}}$.

Let $\rho_\tau(D, f_\tau)$ be the pinball loss of $f_\tau$ on $D$, i.e. $\rho_\tau(D, f_\tau) = \sum_{i=1}^{N} \rho_\tau(y_i, f_\tau(x_i))$.

Then,

$$\rho_\tau(D, f_\tau) = \sum_{i=1}^{N} \rho_\tau(y_i, f_\tau(x_i)) = \sum_{(x_i,y_i)\in S_f^{\text{under}}} (f_\tau(x_i) - y_i)(1 - \tau)$$
$$+ \sum_{(x_i,y_i)\in S_f^{\text{over}}} (f_\tau(x_i) - y_i)(-\tau).$$

We can construct another quantile model $g$, s.t. its prediction for $\tau$, $g_\tau$, is as follows: take any point $(x_k,y_k) \in S_f^{\text{over}}$ and set $g_\tau(x_k) = y_k - \frac{\tau}{2(1-\tau)}(f_\tau(x_k) - y_k)$. For all other points, $(x_i,y_i)_{i \ne k} \in D$, set $g_\tau(x_i) = f_\tau(x_i)$.

Since $y_k - g_\tau(x_k) = \frac{\tau}{2(1-\tau)}(f_\tau(x_k) - y_k) < 0$, we have that $y_k < g_\tau(x_k)$.

Therefore, $|\{(x_i,y_i) \mid y_i \le g_\tau(x_i)\}| = (\tau * N) + 1$ and $|\{(x_i,y_i) \mid y_i > g_\tau(x_i)\}| = ((1-\tau)*N) - 1$.

Therefore, $|\hat{p}_{avg}^{obs}(D,\tau)$ of $g - \tau| > |\hat{p}_{avg}^{obs}(D,\tau)$ of $f - \tau|$, which implies that according to $\{p_j\}_{j=1}^{m}$, $ECE(D,g) > ECE(D,f)$.

We further consider the pinball loss of $g_\tau$ on $D$, $\rho_\tau(D, g_\tau)$.

$\rho_\tau(D, g_\tau) = \sum_{i=1}^{N} \rho_\tau(y_i, g_\tau(x_i)) = \rho_\tau(y_k, g_\tau(x_k)) + \sum_{i \in [N], i \ne k} \rho_\tau(y_i, f_\tau(x_i))$.

Note that the term, $\rho_\tau(y_k, g_\tau(x_k))$ satisfies:

$$
\begin{aligned}
\rho_\tau(y_k, g_\tau(x_k)) &= (g_\tau(x_k) - y_k)(1 - \tau) \\
&= \frac{\tau}{2(1-\tau)}(y_k - f_\tau(x_k))(1 - \tau) \\
&= \frac{\tau}{2}(y_k - f_\tau(x_k)) \\
&< \tau(y_k - f_\tau(x_k)) \\
&= (-\tau)(f_\tau(x_k) - y_k) \\
&= \rho_\tau(y_k, f_\tau(x_k))
\end{aligned}
$$

Therefore, $\rho_\tau(D, g_\tau) < \rho_\tau(D, f_\tau)$, i.e. the pinball loss of $g_\tau$ on $D$ is less than the pinball loss of $f_\tau$ on $D$.

Note that for any $\tau \notin \{p_j\}_{j=1}^m$, we can follow the same steps to construct $g_\tau$ s.t. $|\hat{p}_{avg}^{obs}(D, \tau)$ of $g - \tau| > |\hat{p}_{avg}^{obs}(D, \tau)$ of $f - \tau|$, i.e. $g_\tau$ is more miscalibrated than $f_\tau$. Therefore, for any quantile level $\tau \in (0, 1)$, $\sum_{i=1}^N \rho_\tau(y_i, g_\tau(x_i)) < \sum_{i=1}^N \rho_\tau(y_i, f_\tau(x_i))$, but $\mathrm{ECE}(D, g) > \mathrm{ECE}(D, f)$.

$\square$

## A.2   Proof of Proposition 2

**Proposition 2**. *For any quantile level $p \in (0, 1)$, the true quantile function $\mathbb{Q}_p$ minimizes the calibration objective, $\mathcal{C}(\hat{\mathbb{Q}}_p, p)$. Further, on a finite dataset $D$, the empirical calibration objective, $\mathcal{C}(D, \hat{\mathbb{Q}}_p, p)$, is minimized by an average calibrated solution on $D$, i.e. when $\hat{p}_{avg}^{obs}(D, p) = p$.*

**Proof:**
Recall the calibration objective for a quantile level $p \in (0, 1)$,

$$
\begin{aligned}
\mathcal{C}(\hat{\mathbb{Q}}_p, p) = &\mathbb{I}\{\hat{p}_p < p\} * \mathbb{E}[Y - \hat{\mathbb{Q}}_p | Y > \hat{\mathbb{Q}}_p] * P(Y > \hat{\mathbb{Q}}_p) \\
&+ \mathbb{I}\{\hat{p}_p > p\} * \mathbb{E}[\hat{\mathbb{Q}}_p - Y | \hat{\mathbb{Q}}_p > Y] * P(\hat{\mathbb{Q}}_p > Y), \text{where } \hat{p}_p = P(Y \le \hat{\mathbb{Q}}_p).
\end{aligned}
$$

For the true quantile function $\mathbb{Q}_p$, $P(Y \le \mathbb{Q}_p) = p$, thus achieves the minimum value of $0$ for $\mathcal{C}(\hat{\mathbb{Q}}_p, p)$, as the two non-negative terms of $\mathcal{C}(\hat{\mathbb{Q}}_p, p)$ are $0$.

Further, recall the empirical calibration objective,

$$
\begin{aligned}
\mathcal{C}(D, \hat{\mathbb{Q}}, p) = &\mathbb{I}\{\hat{p}_{avg}^{obs} < p\} * \frac{1}{N} \sum_{i=1}^N \left[ (y_i - \hat{\mathbb{Q}}_p(x_i))\mathbb{I}\{y_i > \hat{\mathbb{Q}}_p(x_i)\} \right] \\
&+ \mathbb{I}\{\hat{p}_{avg}^{obs} > p\} * \frac{1}{N} \sum_{i=1}^N \left[ (\hat{\mathbb{Q}}_p(x_i) - y_i)\mathbb{I}\{\hat{\mathbb{Q}}_p(x_i) > y_i\} \right]
\end{aligned}
$$

An average calibrated solution on the dataset $D$ satisfies $\hat{p}_{avg}^{obs} = p$, thus achieves the minimum value of $0$ for $\mathcal{C}(D, \hat{\mathbb{Q}}, p)$, as the two non-negative terms of $\mathcal{C}(D, \hat{\mathbb{Q}}, p)$ are $0$.

$\square$

## A.3   Derivation of Gradients of Calibration Objective

Denote the CDF and PDF of the random variable $Y$ as $\mathbb{F}_Y$ and $f_Y$.

- When $\hat{p}_p = P(Y \leq \hat{\mathbb{Q}}_p) < p$:

$$\mathcal{C}(\hat{\mathbb{Q}}_p, p) = \mathbb{E}[Y - \hat{\mathbb{Q}}_p | Y > \hat{\mathbb{Q}}_p] * P(Y > \hat{\mathbb{Q}}_p)$$
$$= \left( \mathbb{E}[Y | Y > \hat{\mathbb{Q}}_p] - \hat{\mathbb{Q}}_p \right) * P(Y > \hat{\mathbb{Q}}_p)$$
$$= \left( \frac{\int_{\hat{\mathbb{Q}}_p}^{\infty} y f_Y(y) dy}{P(Y > \hat{\mathbb{Q}}_p)} - \hat{\mathbb{Q}}_p \right) * P(Y > \hat{\mathbb{Q}}_p)$$
$$= \left( \int_{\hat{\mathbb{Q}}_p}^{\infty} y f_Y(y) dy \right) - \left( \hat{\mathbb{Q}}_p * P(Y > \hat{\mathbb{Q}}_p) \right)$$
$$= \left( \int_{\hat{\mathbb{Q}}_p}^{\infty} y f_Y(y) dy \right) - \left( \hat{\mathbb{Q}}_p * (1 - F_Y(\hat{\mathbb{Q}}_p)) \right)$$

Note that,

$$\frac{\partial \left( \int_{\hat{\mathbb{Q}}_p}^{\infty} y f_Y(y) dy \right)}{\partial \hat{\mathbb{Q}}_p} = -\hat{\mathbb{Q}}_p * f_Y(\hat{\mathbb{Q}}_p)$$

$$\frac{\partial \left( \hat{\mathbb{Q}}_p * (1 - F_Y(\hat{\mathbb{Q}}_p)) \right)}{\partial \hat{\mathbb{Q}}_p} = (1 - F_Y(\hat{\mathbb{Q}}_p) + \hat{\mathbb{Q}}_P * (-f_y(\hat{\mathbb{Q}}_p))$$

Therefore,

$$\frac{\partial \mathcal{C}(\hat{\mathbb{Q}}_p, p)}{\partial \hat{\mathbb{Q}}_p} = \frac{\partial \left( \int_{\hat{\mathbb{Q}}_p}^{\infty} y f_Y(y) dy \right)}{\partial \hat{\mathbb{Q}}_p} - \frac{\partial \left( \hat{\mathbb{Q}}_p * (1 - F_Y(\hat{\mathbb{Q}}_p)) \right)}{\partial \hat{\mathbb{Q}}_p}$$
$$= -\hat{\mathbb{Q}}_p * f_Y(\hat{\mathbb{Q}}_p) - (1 - F_Y(\hat{\mathbb{Q}}_p) + \hat{\mathbb{Q}}_P * f_y(\hat{\mathbb{Q}}_p)$$
$$= -(1 - F_Y(\hat{\mathbb{Q}}_p)$$
$$= -P(Y > \hat{\mathbb{Q}}_p)$$

- When $\hat{p}_p = P(Y \leq \hat{\mathbb{Q}}_p) > p$:

$$\mathcal{C}(\hat{\mathbb{Q}}_p, p) = \mathbb{E}[\hat{\mathbb{Q}}_p - Y | \hat{\mathbb{Q}}_p > Y] * P(\hat{\mathbb{Q}}_p > Y)$$
$$= \left( \hat{\mathbb{Q}}_p - \mathbb{E}[Y | \hat{\mathbb{Q}}_p > Y] \right) * P(\hat{\mathbb{Q}}_p > Y)$$
$$= \left( \hat{\mathbb{Q}}_p - \frac{\int_{-\infty}^{\hat{\mathbb{Q}}_p} y f_Y(y) dy}{P(\hat{\mathbb{Q}}_p > Y)} \right) * P(\hat{\mathbb{Q}}_p > Y)$$
$$= \left( \hat{\mathbb{Q}}_p * P(\hat{\mathbb{Q}}_p > Y) \right) - \left( \int_{-\infty}^{\hat{\mathbb{Q}}_p} y f_Y(y) dy \right)$$
$$= \left( \hat{\mathbb{Q}}_p * F_Y(\hat{\mathbb{Q}}_p) \right) - \left( \int_{-\infty}^{\hat{\mathbb{Q}}_p} y f_Y(y) dy \right)$$

Note that,

$$\frac{\partial \left( \hat{\mathbb{Q}}_p * F_Y(\hat{\mathbb{Q}}_p) \right)}{\partial \hat{\mathbb{Q}}_p} = F_Y(\hat{\mathbb{Q}}_p) + \hat{\mathbb{Q}}_P * f_Y(\hat{\mathbb{Q}}_p)$$

$$\frac{\partial \left( \int_{-\infty}^{\hat{\mathbb{Q}}_p} y f_Y(y) dy \right)}{\partial \hat{\mathbb{Q}}_p} = \hat{\mathbb{Q}}_p * f_Y(\hat{\mathbb{Q}}_p)$$

Therefore,

$$
\begin{aligned}
\frac{\partial \mathcal{C}(\hat{\mathbb{Q}}_p, p)}{\partial \hat{\mathbb{Q}}_p} &= \frac{\partial \left( \hat{\mathbb{Q}}_p * F_Y(\hat{\mathbb{Q}}_p) \right)}{\partial \hat{\mathbb{Q}}_p} - \frac{\partial \left( \int_{-\infty}^{\hat{\mathbb{Q}}_p} y f_Y(y) dy \right)}{\partial \hat{\mathbb{Q}}_p} \\
&= F_Y(\hat{\mathbb{Q}}_p) + \hat{\mathbb{Q}}_P * f_Y(\hat{\mathbb{Q}}_p) - \hat{\mathbb{Q}}_p * f_Y(\hat{\mathbb{Q}}_p) \\
&= F_Y(\hat{\mathbb{Q}}_p) \\
&= P(Y < \hat{\mathbb{Q}}_p).
\end{aligned}
$$

$\square$

## A.4 Optimum of Interval Score

Following notation from Section 3.3, we denote $\hat{l} = \hat{\mathbb{Q}}(x, \frac{\alpha}{2})$ and $\hat{u} = \hat{\mathbb{Q}}(x, 1 - \frac{\alpha}{2})$, and we omit conditioning on $x$ for clarity.

Assume $\hat{l} \leq \hat{u}$. Then,

$$
\begin{aligned}
& \mathbb{E}\left[ S_\alpha(\hat{l}, \hat{u}; y) \right] \\
&= \int_{-\infty}^{\hat{l}} S_\alpha(\hat{l}, \hat{u}; y) d\mathbb{F}(y) + \int_{\hat{l}}^{\hat{u}} S_\alpha(\hat{l}, \hat{u}; y) d\mathbb{F}(y) + \int_{\hat{u}}^{\infty} S_\alpha(\hat{l}, \hat{u}; y) d\mathbb{F}(y) \\
&= (\hat{u} - \hat{l}) + \frac{2}{\alpha} \int_{-\infty}^{\hat{l}} (\hat{l} - y) d\mathbb{F}(y) + \frac{2}{\alpha} \int_{\hat{u}}^{\infty} (y - \hat{u}) d\mathbb{F}(y) \\
& \frac{\partial \mathbb{E}\left[ S_\alpha(\hat{l}, \hat{u}; y) \right]}{\partial \hat{l}} = -1 + \frac{2}{\alpha} \int_{-\infty}^{\hat{l}} d\mathbb{F}(y) = -1 + \frac{2}{\alpha} \mathbb{F}(\hat{l}) \\
& \frac{\partial \mathbb{E}\left[ S_\alpha(\hat{l}, \hat{u}; y) \right]}{\partial \hat{u}} = 1 - \frac{2}{\alpha} \int_{\hat{u}}^{\infty} d\mathbb{F}(y) = 1 - \frac{2}{\alpha}(1 - \mathbb{F}(\hat{u})).
\end{aligned}
$$

Setting $\frac{\partial \mathbb{E}\left[ S_\alpha(\hat{l}, \hat{u}; y) \right]}{\partial \hat{l}}$ and $\frac{\partial \mathbb{E}\left[ S_\alpha(\hat{l}, \hat{u}; y) \right]}{\partial \hat{u}}$ to zero reveals the interval loss minima at the respective true quantiles,

$$
\mathbb{F}(\hat{l}) = \frac{\alpha}{2} \text{ and } \mathbb{F}(\hat{u}) = 1 - \frac{\alpha}{2}
$$

$$
\text{i.e. } \hat{l} = \mathbb{Q}(\cdot, \frac{\alpha}{2}) \text{ and } \hat{u} = \mathbb{Q}(\cdot, 1 - \frac{\alpha}{2})
$$

$\square$

# B   Model Agnostic Quantile Regression

## B.1   General Algorithm for Model Agnostic Quantile Regression

As stated in Section 3.1, Algorithm 1 is one implementation of a general model-agnostic quantile regression procedure, in which we take direct estimates of the target density and regress onto these estimates. This general framework is stated in Algorithm 3

---

**Algorithm 3** General Algorithm for Model Agnostic Quantile Regression

---

1: **Input:** Train data $\{x_i, y_i\}_{i=1}^N$
2: Initialize $D \leftarrow \varnothing$
3: **for** $i = 1$ **to** $N$ **do**
4:     Select a set of quantile levels $\{p_k\}_{k=1}^m$, $p_k \in [0, 1]$
5:     $\hat{q}_{i,p_k} \leftarrow$ KDE estimate of $\mathbb{Q}(x_i, p_k)$, $k = 1, \ldots, m$
6:     $D \leftarrow D \cup \{x_i, p_k, \hat{q}_{i,p_k}\}_{k=1}^m$
7: **end for**
8: Use $D$ to fit a regression model $\hat{\mathbb{Q}}$
$$\hat{\mathbb{Q}} : (x_i, p_k) \mapsto \hat{q}_{i,p_k}, \quad k = 1, \ldots, m$$
9: **Output:** $\hat{g}$, $\quad k = 1, \ldots, m$

---

Algorithm 1 implements the KDE step of Algorithm 3 (Line 5 of Algorithm 3) by using a uniform kernel over $\mathcal{X}$ (Line 2 of Algorithm 2) and $\mathcal{Y}$ (Lines 3,5,6 of Algorithm 2).

It should also be noted that many other conditional KDE methods can be used to construct the dataset $D$. We refer the reader to Holmes et al. [25], Hyndman et al. [26] for a more thorough treatment of methods in conditional KDE.

Lastly, this algorithm is model agnostic because *any* regression model can be used for $\hat{g}$ in Algorithm 3, and for $\hat{f}$ and $\hat{g}$ in Algorithm 1. In our specific implementation of Algorithm 1, we used a neural network for $\hat{g}$ to fit the quantile dataset $D$, but we can also use other models, such as a random forest or gradient-boosted trees. In particular, we have replaced $\hat{g}$ in Algorithm 1 with a gradient-boosted tree model and observed very similar UQ performance on the UCI datasets as reported in Section 4.1 (we omit numerical values because they are very similar and otherwise uninformative).

## B.2   Algorithm Complexity

Lines 2 and 3 of Algorithm 2 requires calculating the distance between $x_k$ and all other $x_i$, $1 \leq i \leq N$, and ordering these distances to construct the empirical CDF. Let the distance calculation between a pair of points take constant $C$ time. Ordering the distance requires sorting the $N$ distances. Hence, Lines 5 and 6 takes $\mathcal{O}(N \log N)$ time.

If $K$ points are in the set $E_{k,d_N}$, Lines 5, 6, 7 of Algorithm 2 are done for each of the $K$ points. We consider the worst case when each set $E_{k,d_N}$ contains all $N$ points, which costs $\mathcal{O}(N)$.

The above two procedures are done for all $N$ points (Line 4 of Algorithm 1), therefore the for loop from Lines 4 to 7 in Algorithm 1 requires $\mathcal{O}(N^2 \log N)$ time. This loop takes into account creating the dataset $D$ for the quantile model. The rest of the algorithm constitutes fitting a regression model with this dataset $D$, which we do not analyze here.

We now discuss the space complexity. In Lines 5, 6, 7 of Algorithm 2, we only draw the quantiles at which a discontinuous step occurs in the constructed empirical CDF. For example, if we constructed an empirical CDF with three equally weighted points, we will only draw the quantiles $[1/3, 2/3, 1]$. Following this procedure, in the worst case, for each of the $N$ points, we will construct a CDF with all $N$ points, and hence draw $N$ quantiles to append to the quantile model dataset $D$. If we consider the space complexity of the mean function and the quantile model to be constant, the algorithm requires $O(N^2)$ space in total.

# C  Details on Datasets and Setup of Main Experiments

## C.1  UCI Experiment Details

Here, we provide details on the setup of the UCI experiments presented in Section 4.1.

For each of the 8 UCI datasets, we split $10\%$ of the data into the test set, and we further split $20\%$ of the remaining $90\%$ of the data for the validation, resulting in a train/validation/test split of proportions $72\%, 18\%, 10\%$. For all tasks and all 8 datasets, the data was preprocessed by centering to zero mean and scaling to unit variance.

We used the same NN architecture across all methods: 2 layers of 64 hidden units with ReLU non-linearities. We used the same learning rate, $1e^{-3}$, and the same batch size, 64, for all methods. For all methods, training was stopped early if the validation loss did not decrease for more than 200 epochs, until a maximum of 10000 epochs. If training was stopped early, the final model was backtracked to the model with lowest validation loss.

*IndvCal* (individual calibration) has one hyperparameter, $\alpha$, which balances the NLL loss and the individual calibration loss in the loss function. We 5-fold cross-validated $\alpha$ in [0.0, 1.0] in 20 equi-spaced intervals based on Pareto optimality in test set NLL and adversarial group calibration. If there were multiple $\alpha$ values that were Pareto optimal, we chose the value that had the best test set adversarial group calibration.

*Cali* (penalized calibration loss) has one hyperparameter, $\lambda$, which balances the calibration loss and sharpness penalty in the loss function. We tuned $\lambda$ according to the same grid as above for $\alpha$, based on the criterion of adversarial group calibration. Note that adversarial group calibration will not always favor lower values of $\lambda$ as $\lambda = 0$ will only target average calibration, which, in the degenerate case, may converge to the marginal distribution of $\mathbb{F}_\mathbf{Y}$. This state will achieve very poor adversarial group calibration.

Group batching was applied to *Interval* (interval score) and *Cali* according to the implementation detailed in Appendix C.3. During training, we alternated between "group batching epochs" and "regular batching epochs" (where batches are drawn uniformly from the training set), and the frequency of group batching epochs was a hyperparameter we tuned with cross-validation in [1, 2, 3, 5, 10, 30, 100], based on the criterion of adversarial group calibration.

*MAQR* has a two-step training process: we first learn a mean model, then construct a quantile dataset $D$, then regress onto this dataset with the quantile model. Both the mean model and the quantile model had the same NN architecture as mentioned above. The mean model was trained with the MSE loss according to the same, aforementioned training procedure. For each UCI dataset, we learned one mean model from the first seed, and re-used this mean model for all other seeds. Using this mean model, we then populated the quantile dataset according to the method outlined in Algorithm 1. Algorithm 1 requires one hyperparameter: the distance threshold in $\mathcal{X}$ space ($d_N$ in Line 2 of Algorithm 2). We tuned this hyperparameter by setting the minimum distance required to include $k$ number of points, on average, in constructing an empirical CDF at each training point. We tuned this parameter with cross-validation using the grid $k \in [10, 20, 30, 40, 50]$. The quantile model was trained according to the same training procedure but with one difference: the batch size was set to 1024 because the quantile dataset $D$ could become very large due to many conditional quantile estimates at each training point.

All methods, for all datasets were repeated with 5 seeds: [0, 1, 2, 3, 4].

## C.2  Fusion Experiment Details

We first describe the fusion dataset from Section 4.1, then the details of the fusion experiment set up.

The fusion dataset was recorded from the DIII-D tokamak in San Diego, CA, USA, and describes the dynamics of plasma during a nuclear fusion reaction within the tokamak. Consent and access to use this dataset was obtained via collaborations with the Princeton Plasma Physics Lab.

While the dataset in its raw format is a time-series of the state variables and action variables, for the purposes of a supervised learning problem to learn the dynamics of plasma, it has been re-structrued into a *(state, action, next state)* format. Therefore, the modeling task at hand is to learn the mapping

| | State Variables |
|---|---|
| aminor | Minor Radius |
| dssdenest | Line Averaged Electron Density |
| efsbetan | Normalized Beta |
| efsli | Internal Inductance |
| efsvolume | Plasma Volume |
| ip | Plasma Curent |
| kappa | Elongation |
| R0 | Major Radius |
| tribot | Bottom Triangularity |
| tritop | Top Triangularity |

| | Action Variables |
|---|---|
| pinj_15l | Co-current Beam 1 Power |
| pinj_15r | Co-current Beam 2 Power |
| pinj_21l | Counter-current Beam 1 Power |
| pinj_21r | Counter-current Beam 2 Power |
| pinj_30l | Co-current Beam 3 Power |
| pinj_30r | Co-current Beam 4 Power |
| pinj_33l | Co-current Beam 5 Power |
| pinj_33r | Co-current Beam 6 Power |

Figure 5: **State and Action Variables for Fusion Dataset.** 10 variables describe the current state of plasma, and the action space is 8 dimensional.

*(state, action)* to *(next state)*, and the UQ task is then to learn the distribution over the next state given the current state and action.

There are 10 plasma state variables that we use both as the input state variables and the target variables. For the action variables, we use the power level of 8 neutral beams, which are a primary means of controlling plasma in a tokamak. These variables are described in Figure 5.

As input to the dynamics model, we model the current state as a 200 millisecond (ms) history window of the 10 state variables and 8 action variables, and we model the current action as a 200ms window into the future of the 8 action variables. The target variables are modeled as the change (or delta) of the state variables 200ms in the future. The state and action features are engineered according to the method used by Fu et al. [16]. Each 200ms window is taken as one "frame", the 200ms window is further divided into 2 "frames" of equal length (100ms each), as well as thirds. Then we calculate the mean, variance, and slope of each state and action variable for each of these frames, and collect these statistics as the features. Hence, each 200ms window for 1 variable is summarized into 18 features (3 statistics per frame, and 6 frames per 200ms window). Since there are 10 state variables and 8 action variables, and since the state window is 200ms long and the action window is 200ms long, the input is a 468 dimensional array (10 state variables + 8 previous action variables + 8 current action variables for a total of 26 input variables, and 18 features per variable). Once these features are created, we centered and scaled the inputs and targets to zero mean and unit variance.

There were a total of 100K training data points, and 10K validation and 10K test data points.

The training and hyperparameter tuning procedures were exactly the same as the UCI experiments (detailed above in Appendix C.1), except for two differences: 1) we have increased the NN capacity to 3 hidden layers of 100 hidden units and 2) the batch size was set to 500.

The fusion experiment was likewise repeated 5 times with the seeds [0, 1, 2, 3, 4].

## C.3 Group Batching Implementation Details

Group batching, as introduced in Section 3.4, is a general procedure in which deliberate subsets of the training data are constructed and batched from during train time. There is no "correct" method to form these subsets, because the main point is to simply *avoid drawing batches only from* $\mathbb{F}_X$. One

could consider thresholding the values of each dimension of the domain and discretizing the subsets according to the thresholds, and also taking unions of these discretizations to form new subsets.

This implementation can be computationally demanding, because for each threshold setting, one has to make sure to choose a subset with sufficiently many points, and iteratively increase or decrease the dimension thresholds if no subset with sufficiently many points can be found. This computational cost increases significantly with dimension.

Our implementation of group batching for all our experiments were as follows: we sort the datapoints according to a single dimension, then take consecutive sets of size equal to the batch size, and use these sets as the batches to take gradient steps over during an epoch. We repeat the above process by cycling through each dimension for sorting. While this process is very simple and inexpensive and only considers a single dimension in constructing the subsets, this group batching scheme has shown to be very effective in our experiments.

## C.4 Calculation of Evaluation Metrics

To measure the calibration metrics (average, adversarial group, centered interval), we discretized the expected probabilities from $0.01$ to $0.99$ in $0.01$ increments (i.e. $0.01, 0.02, \ldots, 0.97, 0.98, 0.99$) and calculated ECE according to this finite discretization.

To measure centered interval calibration, for each expected probability $p$, we predict centered $100 \times p\%$ PIs, and calculate $\hat{p}_{avg}^{\text{obs}}$ as the proportion of test points falling within the PI, i.e. $\hat{p}_{avg}^{\text{obs}}(p)$ here would be calculated as

$$\hat{p}_{avg}^{\text{obs}}(p) \text{ for centered intervals } = \frac{1}{N} \sum_{i=1}^{N} \mathbb{I}\{\hat{\mathbb{Q}}_{(0.5-p/2)}(x_i) \leq y_i \leq \hat{\mathbb{Q}}_{(0.5+p/2)}(x_i)\}.$$

The procedure in which we measure adversarial group calibration is the following. For a given test set, we scale group size between $1\%$ and $100\%$ of the full test set size, in 10 equi-spaced intervals, and for each group size, we draw 20 random groups from the test set and record the worst calibration incurred across these 20 random groups. This is also the method used by Zhao et al. [65] to measure adversarial group calibration.

Sharpness was measured as the mean width of the $95\%$ centered PI (i.e. between $p = 0.025$ and $0.975$).

The proper scoring rules (check score, interval score) were measured as the average of the score on the test set.

## C.5 Discussion of Compute and Resources

All experiments were run on a single NVIDIA GeForce GTX 1080Ti GPU, with a Intel(R) Xeon(R) Silver 4110 CPU. The fusion datasets were the largest (100K training) and highest dimensional (468 dimensional) and took the longest to run: training one model to convergence took roughly $\sim 1$ hour.

# D  Full Experimental Results

In this section, we provide all of the experimental results from Section 4, for all metrics: 1) average calibration - sharpness, 2) adversarial group calibration, 3) check score, 4) interval score, and 5) centered interval calibration

For average calibration and sharpness, we present the numeric tables again, along with a visualization which plots calibration and sharpness along the $x$ and $y$ axes.

## D.1  Full UCI Experiment Results

|  |  | *SQR* | *mPAIC* | *Interval* | *Cali* | *MAQR* |
|---|---|---|---|---|---|---|
| UCI | Concrete | $9.3 \pm 1.5(\underline{7.0} \pm 1.0)$ | $6.2 \pm 0.5(14.2 \pm 0.8)$ | $\mathbf{3.7} \pm 0.6(18.1 \pm 0.6)$ | $5.6 \pm 0.8(17.3 \pm 1.5)$ | $5.3 \pm 0.4(16.0 \pm 0.4)$ |
|  | Power | $2.6 \pm 0.4(13.4 \pm 0.2)$ | $5.2 \pm 0.4(13.5 \pm 0.3)$ | $2.2 \pm 0.4(21.0 \pm 1.0)$ | $2.0 \pm 0.1(\underline{13.1} \pm 0.1)$ | $\mathbf{1.6} \pm 0.3(19.9 \pm 0.2)$ |
|  | Wine | $4.2 \pm 0.2(29.5 \pm 0.4)$ | $10.3 \pm 0.3(37.7 \pm 0.5)$ | $5.0 \pm 0.8(41.4 \pm 2.5)$ | $4.2 \pm 0.4(\underline{26.0} \pm 0.8)$ | $\mathbf{2.7} \pm 0.2(39.3 \pm 0.5)$ |
|  | Yacht | $9.4 \pm 0.9(\underline{1.0} \pm 0.1)$ | $10.8 \pm 2.3(2.6 \pm 0.4)$ | $7.5 \pm 0.9(4.5 \pm 1.0)$ | $8.3 \pm 0.6(2.0 \pm 0.4)$ | $\mathbf{6.8} \pm 2.1(2.4 \pm 0.3)$ |
|  | Naval | $9.7 \pm 1.6(3.5 \pm 0.4)$ | $3.1 \pm 0.5(63.0 \pm 1.8)$ | $4.7 \pm 1.4(28.4 \pm 3.6)$ | $5.9 \pm 0.7(3.0 \pm 0.2)$ | $\mathbf{2.3} \pm 0.2(\underline{1.7} \pm 0.1)$ |
|  | Energy | $9.8 \pm 0.8(\underline{2.0} \pm 0.1)$ | $10.4 \pm 0.5(4.3 \pm 0.2)$ | $4.3 \pm 0.6(5.1 \pm 0.9)$ | $5.8 \pm 0.4(3.6 \pm 0.3)$ | $\mathbf{3.5} \pm 1.0(3.2 \pm 0.1)$ |
|  | Boston | $9.0 \pm 0.8(\underline{9.3} \pm 0.7)$ | $8.7 \pm 1.3(12.3 \pm 0.7)$ | $6.9 \pm 1.1(20.3 \pm 0.5)$ | $8.5 \pm 1.5(10.9 \pm 0.6)$ | $\mathbf{6.2} \pm 1.8(10.9 \pm 0.8)$ |
|  | Kin8nm | $4.4 \pm 0.1(\underline{11.4} \pm 0.2)$ | $6.6 \pm 0.4(17.0 \pm 0.5)$ | $2.9 \pm 0.4(16.9 \pm 0.5)$ | $3.5 \pm 0.3(13.7 \pm 0.7)$ | $\mathbf{1.8} \pm 0.4(17.1 \pm 0.1)$ |

Figure 6: **UCI Average Calibration-Sharpness Table.** The table shows mean average calibration (measured by ECE) and sharpness in parentheses, along with $\pm 1$ standard error. The best mean ECE for each dataset has been bolded and the best mean sharpness has been underlined. All values have been multiplied by 100 for readability (same table as Figure 2 (Top), repeated here for completeness).

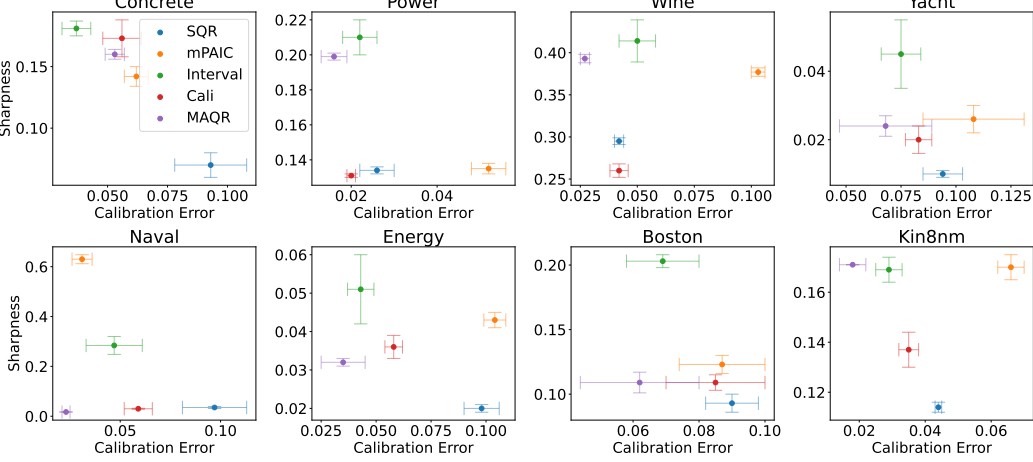

Figure 7: **UCI Average Calibration-Sharpness Plot.** Visualization of average calibration-sharpness from UCI experiments in Section 4.1

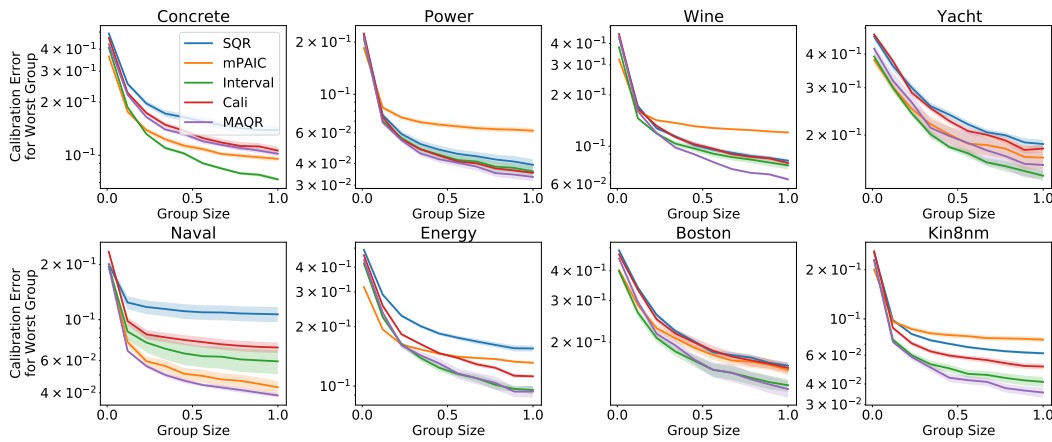

Figure 8:  **UCI Adversarial Group Caibration.**  The plot displays the worst calibration error incurred for any group of any given size. The mean is plotted along with $\pm 1$ standard error in shades. Group size here refers to the proportion of the test dataset size (full results for Figure 2 (Bottom)).

|  | *SQR* | *mPAIC* | *Interval* | *Cali* | *MAQR* |
|---|---|---|---|---|---|
| concrete | $0.085(0.006)$ | $0.085 \pm 0.005$ | $0.086 \pm 0.004$ | $0.118 \pm 0.006$ | $\mathbf{0.059 \pm 0.008}$ |
| power | $\mathbf{0.057 \pm 0.001}$ | $0.070 \pm 0.001$ | $0.062 \pm 0.001$ | $0.064 \pm 0.001$ | $0.058 \pm 0.001$ |
| wine | $0.205 \pm 0.008$ | $0.219 \pm 0.004$ | $0.214 \pm 0.006$ | $0.210 \pm 0.008$ | $\mathbf{0.191 \pm 0.003}$ |
| yacht | $0.012 \pm 0.002$ | $0.015 \pm 0.002$ | $0.018 \pm 0.003$ | $0.019 \pm 0.004$ | $\mathbf{0.007 \pm 0.001}$ |
| naval | $0.070 \pm 0.001$ | $0.276 \pm 0.004$ | $0.066 \pm 0.013$ | $0.159 \pm 0.029$ | $\mathbf{0.004 \pm 0.000}$ |
| energy | $0.014 \pm 0.000$ | $0.015 \pm 0.001$ | $0.017 \pm 0.003$ | $0.017 \pm 0.002$ | $\mathbf{0.010 \pm 0.001}$ |
| boston | $0.088 \pm 0.008$ | $0.095 \pm 0.008$ | $0.094 \pm 0.009$ | $0.103 \pm 0.013$ | $\mathbf{0.063 \pm 0.016}$ |
| kin8nm | $0.078 \pm 0.001$ | $0.104 \pm 0.003$ | $0.077 \pm 0.001$ | $0.096 \pm 0.005$ | $\mathbf{0.070 \pm 0.001}$ |

Figure 9: **UCI Check Score.** Full check score results of UCI experiments from Section 4.1. Mean score across 5 trials is given, along with $\pm 1$ standard error. The best mean has been bolded. *MAQR* tends to achieve the best check score, which is surprising given that *SQR* utilizes the same model class to optimize the check score directly.

|  | *SQR* | *mPAIC* | *Interval* | *Cali* | *MAQR* |
|---|---|---|---|---|---|
| concrete | $2.038 \pm 0.225$ | $1.157 \pm 0.069$ | $0.943 \pm 0.053$ | $1.465 \pm 0.086$ | $\mathbf{0.672 \pm 0.118}$ |
| power | $0.834 \pm 0.022$ | $0.917 \pm 0.021$ | $0.620 \pm 0.010$ | $0.699 \pm 0.019$ | $\mathbf{0.592 \pm 0.009}$ |
| wine | $3.242 \pm 0.166$ | $3.168 \pm 0.019$ | $2.197 \pm 0.045$ | $2.498 \pm 0.135$ | $\mathbf{2.052 \pm 0.052}$ |
| yacht | $0.314 \pm 0.061$ | $0.197 \pm 0.036$ | $0.190 \pm 0.021$ | $0.298 \pm 0.063$ | $\mathbf{0.086 \pm 0.016}$ |
| naval | $0.097 \pm 0.011$ | $3.112 \pm 0.053$ | $0.620 \pm 0.114$ | $1.560 \pm 0.268$ | $\mathbf{0.044 \pm 0.001}$ |
| energy | $0.290 \pm 0.016$ | $0.223 \pm 0.017$ | $0.182 \pm 0.026$ | $0.204 \pm 0.018$ | $\mathbf{0.101 \pm 0.006}$ |
| boston | $1.833 \pm 0.299$ | $1.395 \pm 0.176$ | $1.010 \pm 0.118$ | $1.449 \pm 0.259$ | $\mathbf{0.864 \pm 0.287}$ |
| kin8nm | $1.241 \pm 0.041$ | $1.347 \pm 0.031$ | $0.776 \pm 0.017$ | $1.121 \pm 0.072$ | $\mathbf{0.691 \pm 0.015}$ |

Figure 10: **UCI Interval Score** Full interval score results of UCI experiments from Section 4.1. Mean score across 5 trials is given, along with $\pm 1$ standard error. The best mean has been bolded. *MAQR* tends to achieve the best interval score, which is surprising given that *Interval* utilizes the same model class to optimize the interval score directly.

|  | *SQR* | *mPAIC* | *Interval* | *Cali* | *MAQR* |
|---|---|---|---|---|---|
| concrete | $0.186 \pm 0.031$ | $0.089 \pm 0.005$ | $0.061 \pm 0.008$ | $0.096 \pm 0.013$ | $\mathbf{0.059 \pm 0.020}$ |
| power | $0.045 \pm 0.004$ | $0.068 \pm 0.008$ | $0.023 \pm 0.003$ | $0.037 \pm 0.002$ | $\mathbf{0.010 \pm 0.002}$ |
| wine | $0.053 \pm 0.006$ | $0.169 \pm 0.008$ | $0.079 \pm 0.014$ | $0.065 \pm 0.007$ | $\mathbf{0.045 \pm 0.005}$ |
| yacht | $0.135 \pm 0.009$ | $0.100 \pm 0.020$ | $0.121 \pm 0.005$ | $0.129 \pm 0.016$ | $\mathbf{0.085 \pm 0.024}$ |
| naval | $0.128 \pm 0.031$ | $0.039 \pm 0.003$ | $0.043 \pm 0.014$ | $0.110 \pm 0.013$ | $\mathbf{0.012 \pm 0.002}$ |
| energy | $0.174 \pm 0.011$ | $0.163 \pm 0.009$ | $0.060 \pm 0.010$ | $0.090 \pm 0.011$ | $\mathbf{0.052 \pm 0.018}$ |
| boston | $0.163 \pm 0.020$ | $\mathbf{0.050 \pm 0.007}$ | $0.079 \pm 0.015$ | $0.138 \pm 0.028$ | $0.092 \pm 0.041$ |
| kin8nm | $0.070 \pm 0.005$ | $0.080 \pm 0.002$ | $0.048 \pm 0.006$ | $0.067 \pm 0.005$ | $\mathbf{0.019 \pm 0.008}$ |

Figure 11: **UCI Centered Interval Calibration** Full centered interval calibration results of UCI experiments from Section 4.1. Mean score across 5 trials is given, along with $\pm 1$ standard error. The best mean has been bolded. *MAQR* tends to achieve the best centered interval calibration.

## D.2   Full Fusion Experiment Results

| | | $SQR$ | $mPAIC$ | ***Interval*** | ***Cali*** |
|---|---|---|---|---|---|
| **Fusion** | aminor | $5.8 \pm 0.9(\underline{1.6} \pm 0.0)$ | $13.5 \pm 0.0(5.3 \pm 0.0)$ | $3.6 \pm 0.7(3.7 \pm 0.1)$ | $\mathbf{2.9} \pm 0.2(2.2 \pm 0.0)$ |
| | betan | $\mathbf{3.1} \pm 0.4(\underline{2.8} \pm 0.1)$ | $9.2 \pm 0.4(5.5 \pm 0.1)$ | $5.1 \pm 0.4(5.7 \pm 0.3)$ | $3.4 \pm 0.5(3.3 \pm 0.2)$ |
| | dssdenest | $4.4 \pm 0.5(\underline{7.2} \pm 0.2)$ | $8.4 \pm 0.1(13.7 \pm 0.1)$ | $\mathbf{2.9} \pm 0.3(13.3 \pm 0.4)$ | $4.1 \pm 0.5(8.9 \pm 0.3)$ |
| | ip | $3.2 \pm 0.3(\underline{2.4} \pm 0.1)$ | $13.5 \pm 0.2(9.0 \pm 0.2)$ | $4.4 \pm 0.3(5.4 \pm 0.1)$ | $\mathbf{2.3} \pm 0.2(3.8 \pm 0.1)$ |
| | kappa | $4.4 \pm 0.6(\underline{2.5} \pm 0.1)$ | $14.8 \pm 0.5(9.0 \pm 0.4)$ | $4.2 \pm 0.4(6.1 \pm 0.2)$ | $\mathbf{3.6} \pm 0.2(3.4 \pm 0.1)$ |
| | li | $4.4 \pm 0.6(\underline{1.0} \pm 0.1)$ | $12.6 \pm 0.6(2.7 \pm 0.1)$ | $3.7 \pm 0.6(2.2 \pm 0.1)$ | $\mathbf{2.9} \pm 0.4(1.3 \pm 0.1)$ |
| | R0 | $5.3 \pm 0.8(\underline{3.3} \pm 0.1)$ | $8.7 \pm 0.4(7.2 \pm 0.3)$ | $\mathbf{3.4} \pm 0.2(6.2 \pm 0.1)$ | $3.6 \pm 0.2(4.0 \pm 0.3)$ |
| | tribot | $4.6 \pm 0.4(\underline{2.4} \pm 0.1)$ | $15.1 \pm 0.7(8.8 \pm 0.7)$ | $\mathbf{3.9} \pm 0.5(6.0 \pm 0.2)$ | $4.6 \pm 0.5(3.0 \pm 0.2)$ |
| | tritop | $5.6 \pm 0.7(\underline{2.7} \pm 0.1)$ | $12.9 \pm 0.7(7.2 \pm 0.7)$ | $3.7 \pm 0.8(6.5 \pm 0.4)$ | $\mathbf{2.5} \pm 0.3(4.4 \pm 0.1)$ |
| | volume | $5.8 \pm 1.2(\underline{0.9} \pm 0.0)$ | $16.9 \pm 1.1(3.9 \pm 0.4)$ | $3.6 \pm 0.3(2.0 \pm 0.1)$ | $\mathbf{2.8} \pm 0.1(1.2 \pm 0.0)$ |

Figure 12: **Fusion Average Calibration-Sharpness Table.** The table shows mean average calibration (measured by ECE) and sharpness in parentheses, along with $\pm 1$ standard error. The best mean ECE for each dataset has been bolded and the best mean sharpness has been underlined. All values have been multiplied by 100 for readability. *Cali* tends to achieve the best average calibration, while *SQR* achieves the sharpest predictions (same table as Figure 3 (Top), repeated here for completeness)

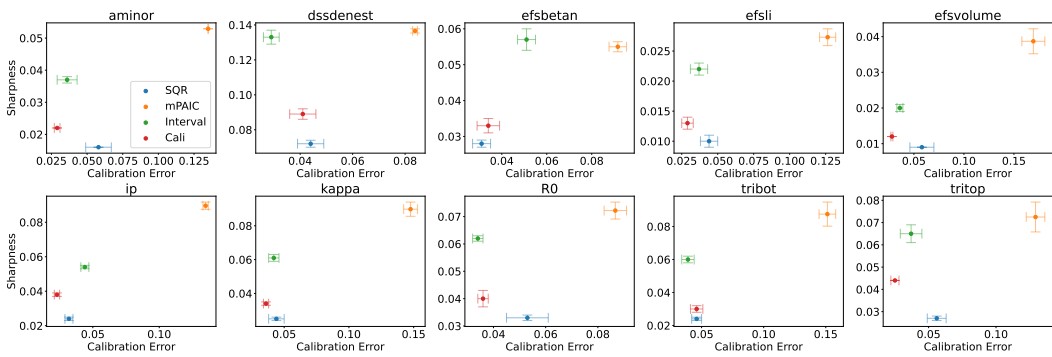

Figure 13: **Fusion Average Calibration-Sharpness Plot.** Visualization of average calibration-sharpness from fusion experiments from Section 4.1
.

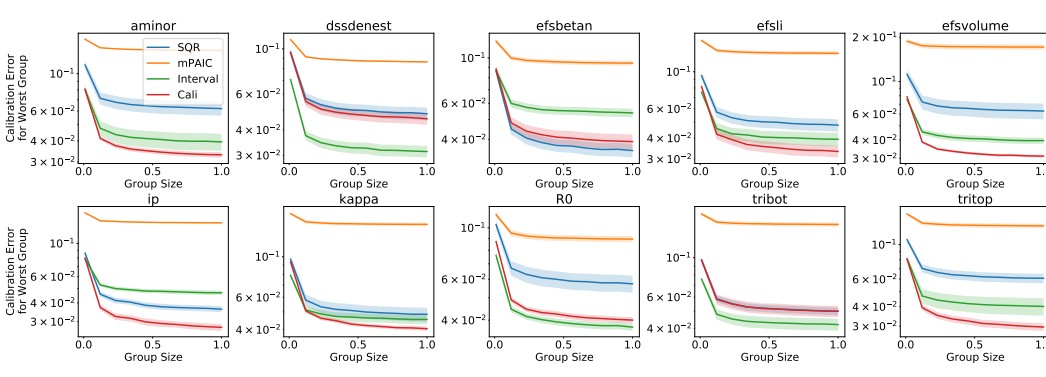

Figure 14: **Fusion Adversarial Group Calibration** Full fusion results of Figure 3 (Bottom). *Cali* and *Interval* tend to achieve the best calibration for any group of any size in the test set.

|  | *SQR* | *mPAIC* | *Interval* | *Cali* |
|---|---|---|---|---|
| aminor | **0.087 ± 0.000** | 0.182 ± 0.000 | 0.097 ± 0.002 | 0.092 ± 0.002 |
| dssdenest | **0.179 ± 0.003** | 0.273 ± 0.003 | 0.180 ± 0.001 | 0.184 ± 0.002 |
| betan | **0.146 ± 0.001** | 0.253 ± 0.005 | 0.153 ± 0.002 | 0.150 ± 0.001 |
| li | **0.097 ± 0.001** | 0.166 ± 0.003 | 0.102 ± 0.001 | 0.104 ± 0.001 |
| volume | **0.051 ± 0.001** | 0.107 ± 0.007 | 0.053 ± 0.001 | 0.052 ± 0.001 |
| ip | **0.068 ± 0.000** | 0.199 ± 0.011 | 0.077 ± 0.002 | 0.076 ± 0.001 |
| kappa | **0.072 ± 0.001** | 0.150 ± 0.004 | 0.079 ± 0.002 | 0.078 ± 0.001 |
| R0 | **0.120 ± 0.001** | 0.208 ± 0.002 | 0.126 ± 0.002 | 0.130 ± 0.005 |
| tribot | **0.084 ± 0.001** | 0.184 ± 0.020 | 0.092 ± 0.001 | 0.096 ± 0.005 |
| tritop | **0.102 ± 0.001** | 0.200 ± 0.018 | 0.107 ± 0.004 | 0.107 ± 0.002 |

Figure 15: **Fusion Check Score** Check score results from fusion experiments in Section 4.1. Mean score across 5 trials is given, along with ±1 standard error. The best mean has been bolded. *SQR* achieves the best check score.

|  | *SQR* | *mPAIC* | *Interval* | *Cali* |
|---|---|---|---|---|
| aminor | 1.181 ± 0.007 | 3.225 ± 0.000 | **1.090 ± 0.017** | 1.207 ± 0.035 |
| dssdenest | 2.387 ± 0.060 | 4.352 ± 0.109 | **1.995 ± 0.011** | 2.369 ± 0.054 |
| betan | 1.970 ± 0.013 | 4.301 ± 0.124 | **1.725 ± 0.021** | 2.002 ± 0.046 |
| li | 1.354 ± 0.018 | 2.923 ± 0.088 | **1.146 ± 0.010** | 1.410 ± 0.015 |
| volume | 0.711 ± 0.023 | 2.036 ± 0.154 | **0.602 ± 0.012** | 0.697 ± 0.026 |
| ip | 0.906 ± 0.005 | 3.203 ± 0.295 | **0.845 ± 0.017** | 0.978 ± 0.026 |
| kappa | 1.010 ± 0.010 | 2.639 ± 0.101 | **0.891 ± 0.015** | 1.066 ± 0.020 |
| R0 | 1.599 ± 0.011 | 3.310 ± 0.064 | **1.414 ± 0.012** | 1.709 ± 0.057 |
| tribot | 1.211 ± 0.009 | 3.185 ± 0.242 | **1.074 ± 0.012** | 1.421 ± 0.115 |
| tritop | 1.481 ± 0.022 | 3.564 ± 0.351 | **1.232 ± 0.034** | 1.444 ± 0.031 |

Figure 16: **Fusion Interval Score** Interval score results from fusion experiments in Section 4.1. Mean score across 5 trials is given, along with ±1 standard error. The best mean has been bolded. *Interval* achieves the best interval score.

|  | *SQR* | *mPAIC* | *Interval* | *Cali* |
|---|---|---|---|---|
| aminor | 0.081 ± 0.006 | 0.268 ± 0.000 | **0.055 ± 0.007** | 0.058 ± 0.004 |
| dssdenest | 0.072 ± 0.003 | 0.162 ± 0.003 | **0.054 ± 0.006** | 0.071 ± 0.008 |
| betan | **0.051 ± 0.001** | 0.160 ± 0.012 | 0.087 ± 0.009 | 0.056 ± 0.007 |
| li | 0.071 ± 0.011 | 0.214 ± 0.017 | 0.069 ± 0.013 | **0.043 ± 0.003** |
| volume | 0.073 ± 0.007 | 0.315 ± 0.022 | 0.063 ± 0.007 | **0.056 ± 0.002** |
| ip | 0.051 ± 0.003 | 0.227 ± 0.020 | 0.077 ± 0.015 | **0.036 ± 0.003** |
| kappa | 0.069 ± 0.008 | 0.254 ± 0.014 | 0.078 ± 0.011 | **0.053 ± 0.004** |
| R0 | 0.068 ± 0.007 | 0.168 ± 0.009 | **0.058 ± 0.007** | 0.066 ± 0.004 |
| tribot | 0.087 ± 0.006 | 0.227 ± 0.016 | **0.074 ± 0.008** | 0.080 ± 0.007 |
| tritop | 0.077 ± 0.006 | 0.231 ± 0.017 | 0.058 ± 0.015 | **0.037 ± 0.004** |

Figure 17: **Fusion Centered Interval Calibration** Full centered interval calibration results from fusion experiments in Section 4.1. Mean score across 5 trials is given, along with ±1 standard error. The best mean has been bolded. *Cali* and *Interval* achieves the best centered interval calibration in 9 out of 10 datasets.

# E  Additional Experiments

## E.1  Details of Synthetic Experiment (Figure 1)

**Dataset**: The synthetic dataset is based on the Boston UCI dataset. A NN model, $\mu_\theta$, was trained on the train split of the Boston dataset to predict the mean by optimizing the MSE loss (same architecture and training details as described in Appendix C.1). Afterwards, all points, $\{(x_i, y_i)\}_{i=1}^N$, in the full Boston dataset were re-labelled with the prediction of the mean model, $(x_i, \mu_\theta(x_i))$. Then, uniform noise, $\epsilon_i$ was added to these re-labelled mean values to create the observations, $\tilde{y}_i$, i.e. $\tilde{y}_i = \mu_\theta(x_i) + \epsilon_i$. The uniform noise was 0 mean, with width of the support equal to 5% of the range of the mean values, i.e $\epsilon_i \sim \text{Unif}[-0.025 * (\max_i y_i - \min_i y_i), 0.025 * (\max_i y_i - \min_i y_i)]$. Thus synthetic dataset is $\{(x_i, \tilde{y}_i)\}_{i=1}^N$.

**Procedure**: The training procedure follows exactly that of the main experiments, which is described in detail in Appendix C.1. The only difference is that the models were trained for the full 1000 epochs, instead of halting training according to the validation loss.

## E.2  Regularizing the Pinball Loss

At first glance, regularization may appear to be the answer to what seems like an overfitting problem with the pinball loss. In fact, regularization has shown to be effective in the "single quantile learning setting", where the quantile level $p$ is fixed and the $p^{\text{th}}$ conditional quantile is learned from data [56, 59]. This setting is concerned with learning $\mathbb{Q}(x)$ for a *given* $p$, which is different from the setting of this work, which considers learning a single model $\mathbb{Q}_p(x)$, which takes both $x$ and $p$ as input and outputs the full predictive distribution by specifying conditional quantiles predictions for all quantiles levels.

In this section, we empirically demonstrate the effect of applying regularization when minimizing the pinball loss to learn $\mathbb{Q}_p(x)$, and show how regularization does not adequately address the issues with the pinball loss.

With the *SQR* method (which optimizes the pinball loss simultaneously for random batches of quantile levels), we applied L1, L2, and dropout, by cross-validating the regularization coefficients in $\{2^i, i \in \text{np.linspace}(-13, 1, 40)\}$ for L1 and L2, and dropout probability $p$ in $\{2^i, i \in \text{np.linspace}(-13, -1, 40)\}$, for the pinball loss criterion (i.e. cross-validate among 40 different regularization coefficients, between $2^{-13}$ and $2^{-1}$ on the log scale). We show the best calibrated regularization result, across all regularization methods and cross-validation (table in same format as Figure 2 in paper).

|          | *SQR*                            | *SQR w/Reg*                      |
|----------|----------------------------------|----------------------------------|
| concrete | $\mathbf{9.3} \pm 1.5(7.0 \pm 1.0)$ | $10.9 \pm 1.0(\underline{6.9} \pm 0.6)$ |
| power    | $2.6 \pm 0.4(\underline{13.4} \pm 0.2)$ | $\mathbf{1.0} \pm 0.1(14.8 \pm 0.1)$ |
| wine     | $\mathbf{4.2} \pm 0.2(29.5 \pm 0.4)$ | $5.1 \pm 0.8(\underline{26.0} \pm 0.5)$ |
| yacht    | $\mathbf{9.4} \pm 0.9(\underline{1.0} \pm 0.1)$ | $12.3 \pm 2.6(\underline{1.0} \pm 0.1)$ |
| naval    | $\mathbf{9.7} \pm 1.6(\underline{3.5} \pm 0.4)$ | $11.5 \pm 2.8(\underline{3.5} \pm 0.3)$ |
| energy   | $9.8 \pm 0.8(2.0 \pm 0.1)$       | $\mathbf{9.4} \pm 1.3(\underline{1.9} \pm 0.2)$ |
| boston   | $\mathbf{9.0} \pm 0.8(9.3 \pm 0.7)$ | $11.6 \pm 1.1(\underline{8.6} \pm 0.8)$ |
| kin8nm   | $4.4 \pm 0.1(11.4 \pm 0.2)$      | $\mathbf{3.5} \pm 0.3(\underline{11.2} \pm 0.2)$ |

Figure 18: **Applying Regularization with SQR: Average Calibration and Sharpness.** The table shows *SQR*'s mean ECE and sharpness (in parentheses) and their standard error with and without regularization. Among the 3 regularization methods (L1, L2, dropout), the method resulting in the best calibration is shown, for each dataset. The best mean ECE for each dataset has been bolded and the best mean sharpness has been underlined. All values have been multiplied by 100 for readability.

Counter-intuitively, regularization tends to further the bias towards sharpness, and upon reflection, this may not be surprising because the range of quantile predictions shrinks: given a quantile model $f : \mathcal{X} \times \mathcal{P} \rightarrow \mathcal{Y}$, where $\mathcal{P}$ is the space of quantile levels in $(0, 1)$, regularization affects the smoothness not only in $\mathcal{X}$, but also in $\mathcal{P}$, hence for any fixed $x \in \mathcal{X}$, the range in predictions for different quantile level inputs also shrinks.

### E.3 Comparison on 95% Prediction Interval Task

In Section 4, we have presented an experiment that is targeted at evaluating the *full predictive distribution* on the 8 UCI datasets.

In this section, we present an experiment that is targeted at constructing a 95% *centered prediction interval (PI)* on the same UCI datasets, for the purposes of comparing against other quantile-based algorithms that are designed to output only a single quantile level. This experiment setup is exactly the same as the prediction intervals experiments in Section 4.1 of the work by Tagasovska and Lopez-Paz [58], and we follow the exact same experiment procedure for a direct comparison against their reported results.

The comparison algorithms here are:

- Dropout [17]: a NN that uses a dropout layer during testing for multiple predictions
- QualityDriven [49]: a NN that optimizes a Binomial likelihood approximation as a surrogate loss for calibration and sharpness
- GradientBoosting [44]: a decision tree based model that optimizes the pinball loss with gradient boosting
- QuantileForest [44]: a decision tree based model that predicts the quantiles based on the trained output of a random forest
- ConditionalGaussian [37]: a probabilistic NN that optimizes the Gaussian NLL to output the parameters of a Gaussian distribution

We show the performance of *MAQR* to represent our proposed methods in this experiment, since *MAQR* performs the best on the full predictive distribution evaluations in the UCI experiments of Section 4.1. We also omit the results of *SQR* Tagasovska and Lopez-Paz [58] in this experiment since we perform a full evaluation comparison with *SQR* in Section 4.1 and Appendix D.1, D.2, and the purpose here is to compare our proposed method against the additional baselines.

In this experiment, since we only output two quantile levels $(0.025, 0.975)$ to construct a single 95% prediction interval, we do not assess calibration (which requires the predictions for all quantile levels) and assess only the observed proportion of test points within the PI (also referred to as "prediction interval coverage probability" or "PICP") and sharpness represented by the width of the PI (also referred to as "mean prediction interval width" or "MPIW"). We refer the reader to Tagasovska and Lopez-Paz [58] for exact details on the experiment setup and the hyperparameters tuned. For *MAQR*, we used the exact same NN architecture (1 layer of 64 hidden units, ReLU non-linearities) as the NN based baselines (*Dropout, QualityDriven, ConditionalGaussian*) and the same training procedure as detailed in Tagasovska and Lopez-Paz [58]. The hyperparameters tuned for *MAQR* are detailed in Appendix C.1.

Summarizing the experimental result in Figure 19:

- Our proposed method (*MAQR*) is capable of consistently producing PIs that have the correct desired coverage (0.95), even in cases when some of the baseline algorithms are not able to.
- Even when the baseline algorithms do produce PIs with correct desired coverage, *MAQR* is able to produce PI's that are much sharper (e.g. mean PI width for naval dataset is an order of magnitude sharper than all other baselines)

On this limited output and evaluation setting, our proposed method is still competitive in its performance. However, it should be noted that this experiment tells *only one facet* of overall UQ quality. Inspecting and evaluating the full predictive uncertainty is necessary for a more thorough evaluation of UQ quality, as done in our main experiments in Section 4.

### E.4 Discussion on Recalibration [35]

The recalibration algorithm by Kuleshov et al. [35] utilizes isotonic regression with a validation set to fine-tune predictive uncertainties from a UQ model. We have applied this recalibration as a post-processing step on the methods presented in Section 4, and the empirical results indicate that its effect on overall improvement in UQ quality is inconclusive. Here, we show its effect on one of

|  | *Dropout* | *QualityDriven* | *GradientBoostingQR* |
|---|---|---|---|
| concrete | none | none | $0.93 \pm 0.00\ (0.71 \pm 0.00)$ |
| power | $0.94 \pm 0.00\ (0.37 \pm 0.00)$ | $0.93 \pm 0.02\ (0.34 \pm 0.19)$ | none |
| wine | none | none | none |
| yacht | $0.97 \pm 0.03\ (0.10 \pm 0.01)$ | $0.92 \pm 0.05\ (0.04 \pm 0.01)$ | $0.95 \pm 0.02\ (0.79 \pm 0.01)$ |
| naval | $0.96 \pm 0.01\ (0.23 \pm 0.00)$ | $0.94 \pm 0.02\ (0.21 \pm 0.11)$ | none |
| energy | $0.91 \pm 0.04\ (0.17 \pm 0.01)$ | $0.91 \pm 0.04\ (0.10 \pm 0.05)$ | none |
| boston | none | none | $0.89 \pm 0.00\ (0.75 \pm 0.00)$ |
| kin8nm | none | $0.96 \pm 0.00\ (0.84 \pm 0.00)$ | none |

|  | *QuantileForest* | *ConditionalGaussian* | *MAQR* |
|---|---|---|---|
| concrete | $0.96 \pm 0.01\ (0.37 \pm 0.02)$ | $0.94 \pm 0.03\ (0.32 \pm 0.09)$ | $0.93 \pm 0.01\ (0.26 \pm 0.01)$ |
| power | $0.94 \pm 0.01\ (0.18 \pm 0.00)$ | $0.94 \pm 0.01\ (0.18 \pm 0.00)$ | $0.95 \pm 0.01\ (0.28 \pm 0.03)$ |
| wine | none | $0.94 \pm 0.02\ (0.49 \pm 0.03)$ | $0.95 \pm 0.02\ (0.56 \pm 0.06)$ |
| yacht | $0.97 \pm 0.04\ (0.28 \pm 0.11)$ | $0.93 \pm 0.06\ (0.03 \pm 0.01)$ | $0.92 \pm 0.03\ (0.03 \pm 0.01)$ |
| naval | $0.92 \pm 0.01\ (0.22 \pm 0.00)$ | $0.96 \pm 0.01\ (0.15 \pm 0.25)$ | $0.94 \pm 0.01\ (0.03 \pm 0.00)$ |
| energy | $0.95 \pm 0.02\ (0.15 \pm 0.01)$ | $0.94 \pm 0.03\ (0.12 \pm 0.18)$ | $0.94 \pm 0.02\ (0.05 \pm 0.01)$ |
| boston | $0.95 \pm 0.03\ (0.37 \pm 0.02)$ | $0.94 \pm 0.03\ (0.55 \pm 0.20)$ | $0.95 \pm 0.02\ (0.34 \pm 0.09)$ |
| kin8nm | none | $0.93 \pm 0.01\ (0.20 \pm 0.01)$ | $0.93 \pm 0.00\ (0.20 \pm 0.01)$ |

Figure 19: **95% PI PICP and MPIW** The test average and standard deviation PICP of models with validation PICP in $[92.5\%, 97.5\%]$ is shown, and the test average and standard deviation MPIW is shown in parantheses. "none" indicates the method could not find a model with validation PICP in $[92.5\%, 97.5\%]$.

our methods, *Interval*, because we observe the same pattern across all the methods, including the baseline algorithms.

Recalibration tends to improve average calibration. This is expected, because recalibration specifically targets average calibration. However, it does so at a cost in sharpness. This is evident in the recalibrated output moving upper left in the average calibration-sharpness plot in Figure 21.

However, there is little to no improvement in adversarial group calibration (except for the Naval dataset) as shown in Figure 22, which seems to indicate that the improvement in average calibration was not meaningful (i.e. the recalibrated result is not closer to individual calibration). This is also the observation made by Zhao et al. [65].

At the same time, the proper scoring rules improved on average (Figures 23, 24), but interval calibration tended to worsen (Figure 25).

Based on these metrics, it is difficult to conclude on whether recalibration by Kuleshov et al. [35] is a beneficial step or not for overall UQ quality. If a practitioner is primarily concerned with average calibration, the results indicate that recalibration *is* a beneficial step, but if converging to the true conditional distribution is the primary objective, recalibration does not seem to be a robust remedy.

|  | *Interval* | *Interval Recalibrated* |
|---|---|---|
| concrete | $3.7 \pm 0.6(\underline{18.1} \pm 0.6)$ | $\mathbf{3.1} \pm 0.4(26.3 \pm 1.8)$ |
| power | $2.2 \pm 0.4(21.0 \pm 1.0)$ | $\mathbf{1.0} \pm 0.1(\underline{20.6} \pm 0.5)$ |
| wine | $5.0 \pm 0.8(\underline{41.4} \pm 2.5)$ | $\mathbf{2.6} \pm 0.2(50.5 \pm 1.8)$ |
| yacht | $7.5 \pm 0.9(\underline{4.5} \pm 1.0)$ | $\mathbf{4.7} \pm 0.5(5.6 \pm 1.1)$ |
| naval | $4.7 \pm 1.4(28.4 \pm 3.6)$ | $\mathbf{1.3} \pm 0.1(\underline{21.9} \pm 3.3)$ |
| energy | $4.3 \pm 0.6(\underline{5.1} \pm 0.9)$ | $\mathbf{3.8} \pm 0.6(6.9 \pm 1.2)$ |
| boston | $6.9 \pm 1.1(\underline{20.3} \pm 0.5)$ | $\mathbf{5.4} \pm 0.9(30.8 \pm 2.6)$ |
| kin8nm | $2.9 \pm 0.4(\underline{16.9} \pm 0.5)$ | $\mathbf{1.1} \pm 0.1(20.6 \pm 0.3)$ |

Figure 20: **UCI Average Calibration-Sharpness Table with Recalibration.** Recalibration tends to trade off sharpness for average calibration. Better mean average calibration has been bolded, and better sharpness has been underlined. All values have been multiplied by 100 for readability.

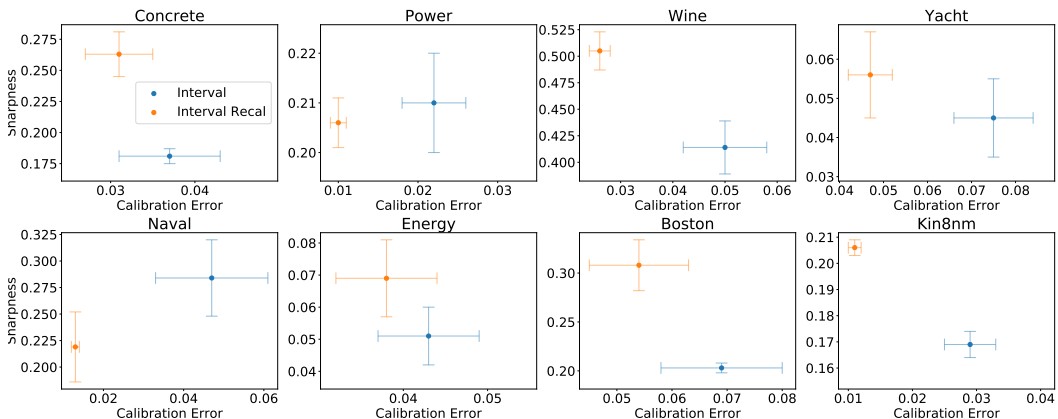

Figure 21: **UCI Average Calibration-Sharpness Plots with Recalibration.** Recalibration tends to trade off sharpness for average calibration. This is evident as the recalibrated predictions move *upper left*.

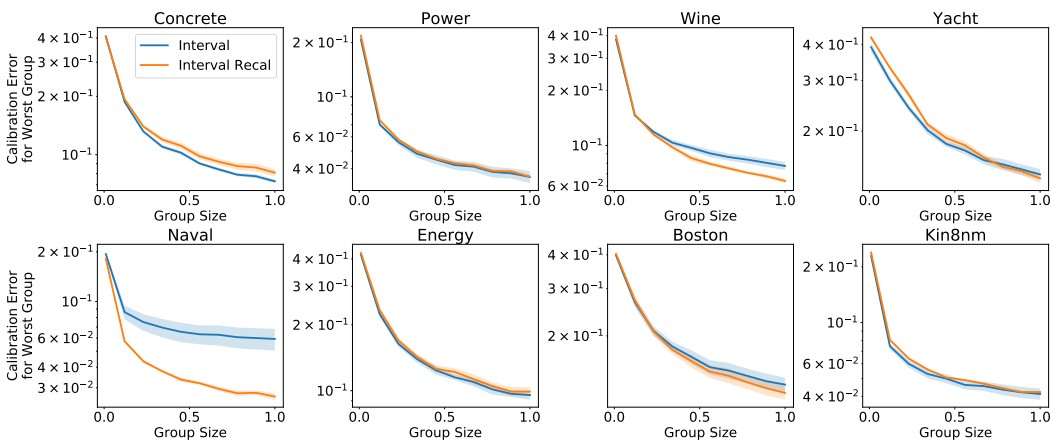

Figure 22: **UCI Adversarial Group Calibration with Recalibration.** Recalibration in general shows little improvement in adversarial group calibration.

|          | *Interval*          | *Interval Recalibrated*    |
| -------- | ------------------- | -------------------------- |
| concrete | $0.086 \pm 0.004$   | $\mathbf{0.077 \pm 0.004}$ |
| power    | $0.062 \pm 0.001$   | $\mathbf{0.061 \pm 0.001}$ |
| wine     | $0.214 \pm 0.006$   | $\mathbf{0.209 \pm 0.013}$ |
| yacht    | $\mathbf{0.018 \pm 0.003}$ | $0.018 \pm 0.004$   |
| naval    | $0.066 \pm 0.013$   | $\mathbf{0.062 \pm 0.012}$ |
| energy   | $0.017 \pm 0.003$   | $\mathbf{0.016 \pm 0.003}$ |
| boston   | $0.094 \pm 0.009$   | $\mathbf{0.076 \pm 0.014}$ |
| kin8nm   | $\mathbf{0.077 \pm 0.001}$ | $\mathbf{0.077 \pm 0.002}$ |

Figure 23: **UCI Check Score with Recalibration.** Recalibration tends to improve the check score.

|          | *Interval*          | *Interval Recalibrated* |
|----------|---------------------|-------------------------|
| concrete | $0.943 \pm 0.053$   | $\mathbf{0.778 \pm 0.064}$ |
| power    | $0.620 \pm 0.010$   | $\mathbf{0.616 \pm 0.006}$ |
| wine     | $2.197 \pm 0.045$   | $\mathbf{1.921 \pm 0.024}$ |
| yacht    | $0.190 \pm 0.021$   | $\mathbf{0.158 \pm 0.025}$ |
| naval    | $\mathbf{3.112 \pm 0.053}$ | $3.150 \pm 0.050$   |
| energy   | $0.182 \pm 0.026$   | $\mathbf{0.148 \pm 0.028}$ |
| boston   | $1.010 \pm 0.118$   | $\mathbf{0.931 \pm 0.107}$ |
| kin8nm   | $0.776 \pm 0.017$   | $\mathbf{0.754 \pm 0.023}$ |

Figure 24: **UCI Interval Score with Recalibration.** Recalibration tends to improve the interval score.

|          | *Interval*          | *Interval Recalibrated* |
|----------|---------------------|-------------------------|
| concrete | $\mathbf{0.061 \pm 0.008}$ | $0.068 \pm 0.015$   |
| power    | $\mathbf{0.023 \pm 0.003}$ | $0.028 \pm 0.008$   |
| wine     | $\mathbf{0.079 \pm 0.014}$ | $\mathbf{0.079 \pm 0.019}$ |
| yacht    | $\mathbf{0.121 \pm 0.005}$ | $0.136 \pm 0.025$   |
| naval    | $\mathbf{0.043 \pm 0.014}$ | $0.105 \pm 0.016$   |
| energy   | $\mathbf{0.060 \pm 0.010}$ | $0.066 \pm 0.005$   |
| boston   | $0.079 \pm 0.015$   | $\mathbf{0.078 \pm 0.012}$ |
| kin8nm   | $\mathbf{0.048 \pm 0.006}$ | $0.061 \pm 0.010$   |

Figure 25: **UCI Centered Interval Calibration with Recalibration.** Recalibration tends to worsen centered interval calibration.

# F  Ablation Study

## F.1  Ablation Study Details

The ablation study from Section 4.2 (with full results in Appendix F.2) investigates the effect of group batching on two methods: *Cali* (combined calibration loss, one of our proposed methods) and *SQR* (a baseline method).

For each method, we applied group batching by tuning the group batching frequency hyperparameter with cross-validation according to the details in Appendix C.1.

When we did not apply group batching, each batch was a uniform draw from the training dataset, which is the default setting in most batch optimization procedures.

## F.2  Full Ablation Study Experiment Results

The full set of results from the ablation study presented in Section 4.2 is provided here. To re-iterate the purpose of this study: we show how group batching affects UQ performance on two methods: *Cali* (combined calibration loss, which is one of our proposed methods) and *SQR* (a baseline method).

We present the effect of group batching via all of the evaluation metrics (average calibration, sharpness, adversarial group calibration, check score, interval score, and centered interval calibration).

|  | *Cali* | |
| --- | --- | --- |
|  | *Random Batch* | *Group Batch* |
| Concrete | $6.6 \pm 0.9 (17.6 \pm 2.3)$ | $\mathbf{5.6} \pm 0.8 (\underline{17.3} \pm 1.5)$ |
| Power | $\mathbf{1.7} \pm 0.2 (14.2 \pm 0.3)$ | $2.0 \pm 0.1 (\underline{13.1} \pm 0.1)$ |
| Wine | $4.4 \pm 0.5 (\underline{25.6} \pm 0.8)$ | $\mathbf{4.2} \pm 0.4 (26.0 \pm 0.8)$ |
| Yacht | $11.1 \pm 1.8 (\underline{1.8} \pm 0.1)$ | $\mathbf{8.3} \pm 0.6 (2.0 \pm 0.4)$ |
| Naval | $2.8 \pm 0.2 (\underline{12.1} \pm 3.1)$ | $\mathbf{2.4} \pm 0.3 (50.6 \pm 8.6)$ |
| Energy | $9.2 \pm 0.3 (\underline{2.8} \pm 0.1)$ | $\mathbf{5.8} \pm 0.4 (3.6 \pm 0.3)$ |
| Boston | $9.7 \pm 1.3 (\underline{10.2} \pm 0.7)$ | $\mathbf{8.5} \pm 1.5 (10.9 \pm 0.6)$ |
| Kin8nm | $\mathbf{3.4} \pm 0.3 (\underline{13.7} \pm 0.4)$ | $3.5 \pm 0.3 (\underline{13.7} \pm 0.7)$ |

|  | *SQR* | |
| --- | --- | --- |
|  | *Random Batch* | *Group Batch* |
| Concrete | $9.8 \pm 1.3 (\underline{7.0} \pm 0.6)$ | $\mathbf{7.1} \pm 0.9 (8.5 \pm 0.6)$ |
| Power | $\mathbf{2.5} \pm 0.3 (14.0 \pm 0.5)$ | $2.9 \pm 0.5 (\underline{13.6} \pm 0.8)$ |
| Wine | $4.5 \pm 0.4 (29.7 \pm 0.5)$ | $\mathbf{4.0} \pm 0.4 (\underline{28.5} \pm 0.8)$ |
| Yacht | $9.0 \pm 0.9 (\underline{0.9} \pm 0.1)$ | $\mathbf{8.9} \pm 0.9 (2.3 \pm 0.2)$ |
| Naval | $8.6 \pm 1.6 (\underline{3.6} \pm 0.1)$ | $\mathbf{5.3} \pm 0.8 (6.0 \pm 0.5)$ |
| Energy | $10.2 \pm 0.8 (\underline{1.8} \pm 0.1)$ | $\mathbf{6.9} \pm 1.1 (2.4 \pm 0.2)$ |
| Boston | $10.9 \pm 1.0 (\underline{8.8} \pm 1.1)$ | $\mathbf{9.8} \pm 1.2 (9.5 \pm 0.9)$ |
| Kin8nm | $4.7 \pm 0.3 (\underline{11.1} \pm 0.1)$ | $\mathbf{3.9} \pm 0.4 (11.3 \pm 0.2)$ |

Figure 26: **Group Batching Ablation: Average Calibration and Sharpness.** The table shows mean ECE and sharpness (in parentheses) and their standard error with and without group batching. The best mean ECE for each dataset has been bolded and the best mean sharpness has been underlined for *Cali* and *SQR* separately. All values have been multiplied by 100 for readability. This is the same table as Figure 4 from Section 4.2, which is repeated here for completeness.

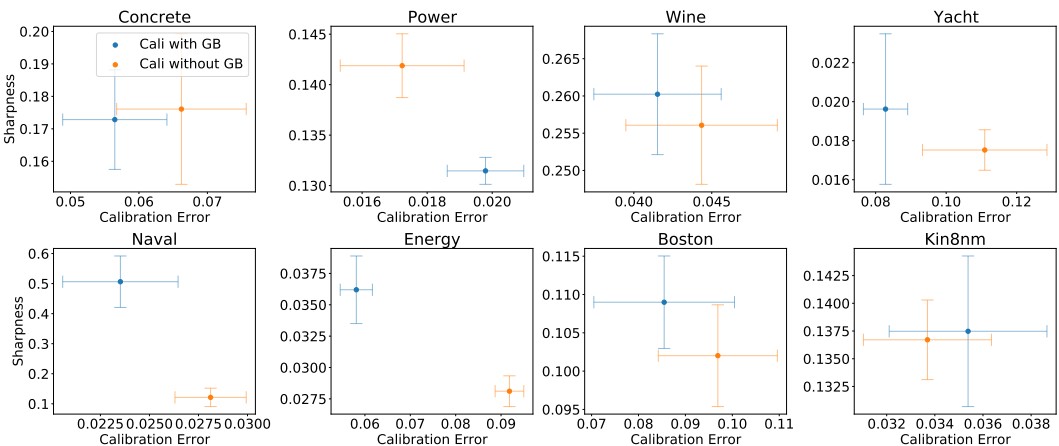

Figure 27: **Group Batching Ablation with *Cali* Method: Average Calibration and Sharpness Plot.** GB in legend refers to group batching. Group batching tends to improve calibration and worsen sharpness for the *Cali* method.

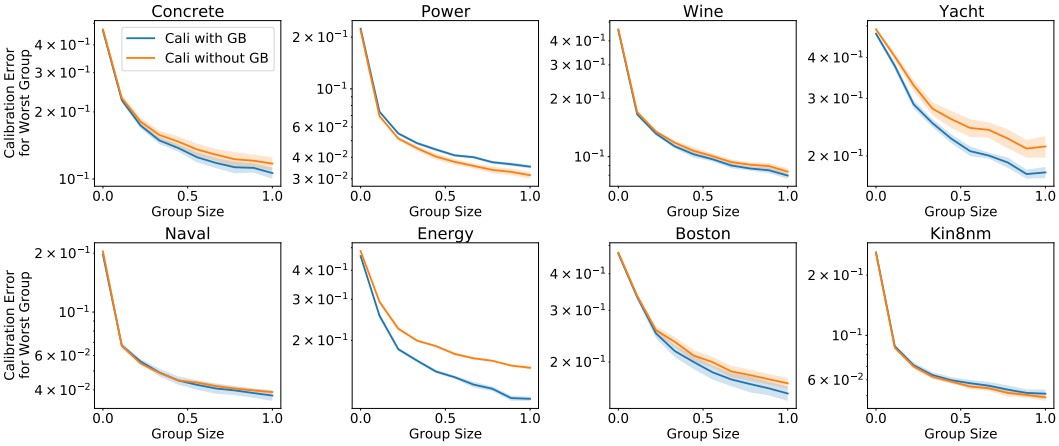

Figure 28: **Group Batching Ablation with *Cali* Method: Adversarial Group Calibration.** GB in legend refers to group batching. Group batching tends to improve adversarial group calibration for the *Cali* method.

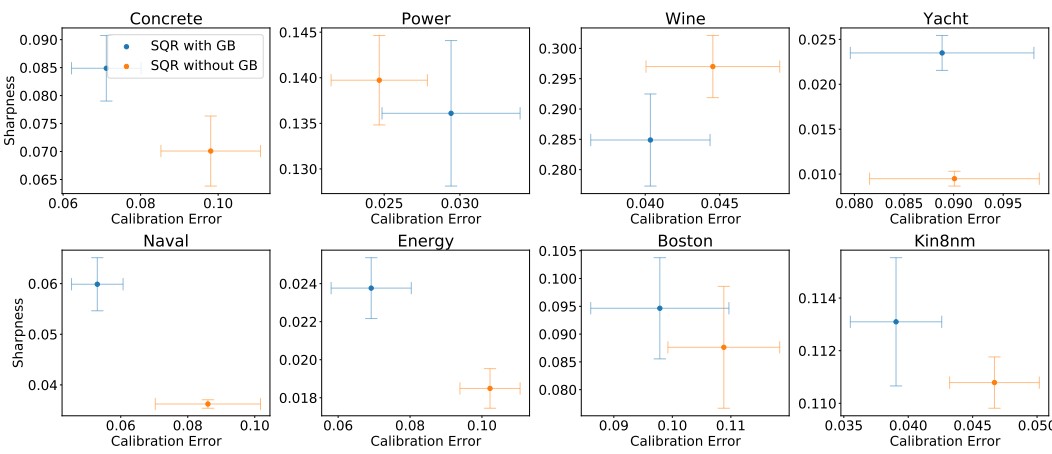

Figure 29: **Group Batching Ablation with *SQR* Method: Average Calibration and Sharpness Plot.** GB in legend refers to group batching. Group batching tends to improve calibration and worsen sharpness for *SQR* method.

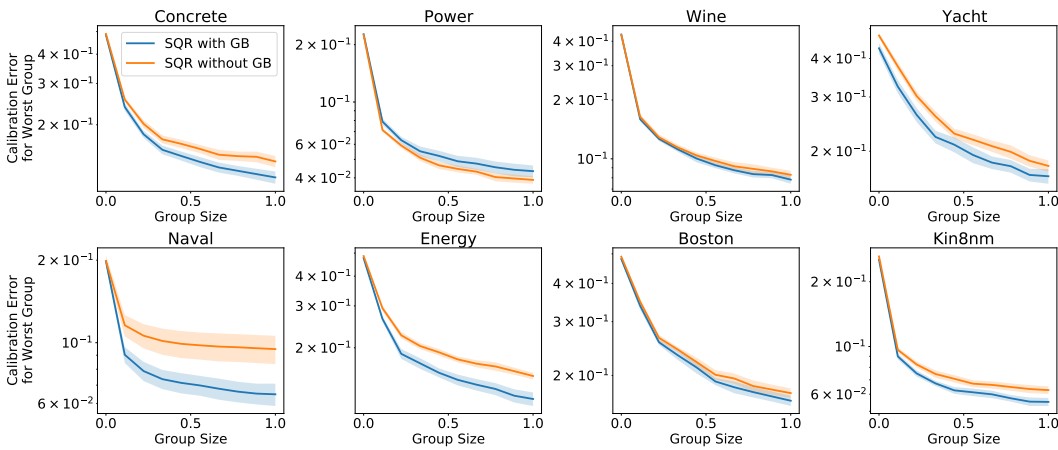

Figure 30: **Group Batching Ablation with *SQR* Method: Adversarial Group Calibration.** GB in legend refers to group batching. Group batching tends to improve adversarial group calibration for *SQR* method.

| | *Cali* | | *SQR* | |
| | Random Batch | Group Batch | Random Batch | Group Batch |
|---|---|---|---|---|
| Concrete | $0.120 \pm 0.007$ | $\mathbf{0.118 \pm 0.006}$ | $0.083 \pm 0.006$ | $\mathbf{0.077 \pm 0.004}$ |
| Power | $\mathbf{0.063 \pm 0.002}$ | $0.064 \pm 0.001$ | $\mathbf{0.056 \pm 0.001}$ | $0.057 \pm 0.001$ |
| Wine | $\mathbf{0.209 \pm 0.007}$ | $0.210 \pm 0.008$ | $\mathbf{0.206 \pm 0.008}$ | $\mathbf{0.206 \pm 0.008}$ |
| Yacht | $\mathbf{0.018 \pm 0.002}$ | $0.019 \pm 0.004$ | $\mathbf{0.011 \pm 0.002}$ | $0.013 \pm 0.002$ |
| Naval | $\mathbf{0.042 \pm 0.009}$ | $0.159 \pm 0.029$ | $\mathbf{0.007 \pm 0.000}$ | $0.014 \pm 0.001$ |
| Energy | $\mathbf{0.016 \pm 0.000}$ | $0.017 \pm 0.002$ | $\mathbf{0.013 \pm 0.000}$ | $\mathbf{0.013 \pm 0.001}$ |
| Boston | $\mathbf{0.100 \pm 0.013}$ | $0.103 \pm 0.013$ | $0.091 \pm 0.007$ | $\mathbf{0.089 \pm 0.008}$ |
| Kin8nm | $\mathbf{0.094 \pm 0.002}$ | $0.096 \pm 0.005$ | $0.079 \pm 0.001$ | $\mathbf{0.077 \pm 0.000}$ |

Figure 31: **Group Batching Ablation: Check Score.** This table shows mean test check score and their standard error with and without group batching for both *Cali* and *SQR*. The best mean for each dataset has been bolded for *Cali* and *SQR* separately. While the general pattern is that group batching worsens the check score, this is expected because group batching tends to worsen sharpness and the check score favors sharpness (Proposition 1 in main paper). Still, the change in the mean of the check score tends to be insignificant when considering the standard errors (except for Naval dataset).

| | Cali | | SQR | |
|---|---|---|---|---|
| | Random Batch | Group Batch | Random Batch | Group Batch |
| Concrete | $1.498 \pm 0.083$ | $\mathbf{1.465 \pm 0.086}$ | $1.254 \pm 0.120$ | $\mathbf{1.079 \pm 0.066}$ |
| Power | $\mathbf{0.667 \pm 0.025}$ | $0.699 \pm 0.019$ | $\mathbf{0.603 \pm 0.016}$ | $0.615 \pm 0.014$ |
| Wine | $\mathbf{2.495 \pm 0.130}$ | $2.498 \pm 0.135$ | $\mathbf{2.325 \pm 0.107}$ | $2.362 \pm 0.117$ |
| Yacht | $\mathbf{0.276 \pm 0.040}$ | $0.298 \pm 0.063$ | $0.177 \pm 0.033$ | $\mathbf{0.164 \pm 0.029}$ |
| Naval | $\mathbf{0.479 \pm 0.098}$ | $1.560 \pm 0.268$ | $\mathbf{0.069 \pm 0.003}$ | $0.144 \pm 0.014$ |
| Energy | $0.218 \pm 0.009$ | $\mathbf{0.204 \pm 0.018}$ | $0.191 \pm 0.006$ | $\mathbf{0.172 \pm 0.012}$ |
| Boston | $\mathbf{1.437 \pm 0.255}$ | $1.449 \pm 0.259$ | $1.284 \pm 0.151$ | $\mathbf{1.217 \pm 0.152}$ |
| Kin8nm | $\mathbf{1.102 \pm 0.031}$ | $1.121 \pm 0.072$ | $0.914 \pm 0.016$ | $\mathbf{0.871 \pm 0.011}$ |

Figure 32: **Group Batching Ablation: Interval Score.** This table shows mean test interval score and their standard error with and without group batching for both *Cali* and *SQR*. The best mean for each dataset has been bolded for *Cali* and *SQR* separately. The general pattern is that the interval score worsens for *Cali* and improves for *SQR*. However, the change tends to be insignificant when considering the standard errors (except for Naval dataset).

| | Cali | | SQR | |
|---|---|---|---|---|
| | Random Batch | Group Batch | Random Batch | Group Batch |
| Concrete | $0.102 \pm 0.014$ | $\mathbf{0.096 \pm 0.013}$ | $0.188 \pm 0.029$ | $\mathbf{0.127 \pm 0.014}$ |
| Power | $\mathbf{0.033 \pm 0.003}$ | $0.037 \pm 0.002$ | $\mathbf{0.040 \pm 0.008}$ | $0.047 \pm 0.006$ |
| Wine | $0.072 \pm 0.007$ | $\mathbf{0.065 \pm 0.007}$ | $0.057 \pm 0.005$ | $\mathbf{0.055 \pm 0.005}$ |
| Yacht | $0.139 \pm 0.018$ | $\mathbf{0.129 \pm 0.016}$ | $0.128 \pm 0.027$ | $\mathbf{0.119 \pm 0.020}$ |
| Naval | $0.048 \pm 0.006$ | $\mathbf{0.034 \pm 0.002}$ | $0.114 \pm 0.032$ | $\mathbf{0.066 \pm 0.009}$ |
| Energy | $0.146 \pm 0.013$ | $\mathbf{0.090 \pm 0.011}$ | $0.171 \pm 0.018$ | $\mathbf{0.104 \pm 0.020}$ |
| Boston | $0.142 \pm 0.032$ | $\mathbf{0.138 \pm 0.028}$ | $0.195 \pm 0.021$ | $\mathbf{0.173 \pm 0.027}$ |
| Kin8nm | $\mathbf{0.061 \pm 0.004}$ | $0.067 \pm 0.005$ | $0.079 \pm 0.003$ | $\mathbf{0.069 \pm 0.009}$ |

Figure 33: **Group Batching Ablation: Centered Interval Calibration.** This table shows mean test centered interval calibration (measured by ECE for centered intervals) score and their standard error with and without group batching for both *Cali* and *SQR*. The best mean for each dataset has been bolded for *Cali* and *SQR* separately. The general pattern is that group batching improves centered interval calibration for both *Cali* and *SQR*. While the change tends to be insignificant when considering the standard errors, the improvement is significant in numerous cases (e.g. Naval and Energy for both methods, Concrete with *SQR*).

# G Considerations for Epistemic Uncertainty

## G.1 Sources for Epistemic Uncertainty

The primary focus of this paper is on learning a quantile model. For any single setting of the parameters of the quantile model, the model outputs the current best estimate of the *true underlying distribution* of the dataset. Following the notation laid out in Section 2, the learned quantile model $\hat{\mathbb{Q}}$ is a best approximation to $\mathbb{Q}$, the quantile function of the true distribution.

Meanwhile, epistemic uncertainty refers to the uncertainty *in making the distributional prediction,* $\hat{\mathbb{Q}}$. Pearce et al. [49] provides one method of decomposing the sources of epistemic uncertainty in a regression setting:

- *Model misspecification:* $\hat{\mathbb{Q}}$ may lack the flexibility to accurately model $\mathbb{Q}$, leading to systematic bias.

- *Data uncertainty:* $\hat{\mathbb{Q}}$ may not be estimated using a representative sample $\{x_i, y_i\}$ from the assumed underlying distribution.

- *Parameter uncertainty:* $\hat{\mathbb{Q}}$ may not be estimated using a large enough quantity of samples, leading to uncertainty about the estimated quantity.

Pearce et al. [49] has argued that, given the rich class of function approximators at hand today (NN, deep trees, ensembles), model misspecification can be ignored, which we believe is reasonable. In modeling the remaining sources of uncertainties in $\hat{\mathbb{Q}}$, we can incorporate common standard methods to quantify epistemic uncertainty, including boostrapping the data, creating an ensemble of estimates for $\hat{\mathbb{Q}}$ with random parameter initializations, or fitting a residual process [38]. Here, we describe one combination of these methods: an ensemble of estimates of the learned quantile function $\{\hat{\mathbb{Q}}^{(1)}, \hat{\mathbb{Q}}^{(2)}, ...\}$ each trained with random initialization (to address parameter uncertainty), on a bootstrapped sample of the training data (to address data uncertainty). The uncertainty over this set of models is the epistemic uncertainty.

## G.2 Expressing and Utilizing Epistemic Uncertainty

Once we decide on methods to quantify the epistemic uncertainty, the next question is *how to express the epistemic uncertainty*, especially when combining it with the current prediction of the aleatoric uncertainty. This is still an open research question, especially in the regression setting, and methods of combining aleatoric with epistemic uncertainty will differ for how the uncertainty is represented (e.g. density estimates, quantiles, prediction intervals).

One method of combination is to consider the utility of quantifying epistemic uncertainty. Intuitively, for a given input, if we have high epistemic uncertainty, the combined uncertainty should be higher (i.e. less confident prediction), and vice versa. If we consider a single quantile, it is unclear whether lower confidence (due to epistemic uncertainty) would equate to a lower or higher quantile estimate. However, if we consider constructing a centered prediction interval for total uncertainty, it is straightforward to see that lower confidence should widen the interval, by raising the upper bound (quantile level above $0.5$) and lowering the lower bound (quantile level below $0.5$). This conservative upper and lower bound can be constructed with the bootstrap distribution of each quantile according to each ensemble member prediction. This is also the method utilized in Pearce et al. [49].

Suppose we have an ensemble of $M$ quantile models: $\{\hat{\mathbb{Q}}^{(1)}, \hat{\mathbb{Q}}^{(2)}, \ldots, \hat{\mathbb{Q}}^{(M)}\}$. For any test point $x^*$ and test coverage level $(1 - \alpha^*)$, the total uncertainty represented by a centered prediction interval

with upper bound $\hat{u}$ and lower bound $\hat{l}$ is constructed as:

$$\hat{l} = \bar{\mu}(x^*, \alpha^*/2) - z\frac{s(x^*, \alpha^*/2)}{\sqrt{M}}$$

$$\hat{u} = \bar{\mu}(x^*, 1 - \alpha^*/2) + z\frac{s(x^*, 1 - \alpha^*/2)}{\sqrt{M}}$$

$$\bar{\mu}(x^*, p^*) = \frac{1}{M}\sum_{i=1}^{M}\hat{\mathbb{Q}}_{p^*}^{(i)}(x^*)$$

$$s(x^*, p^*) = \sqrt{\frac{1}{M-1}\sum_{i=1}^{M}(\hat{\mathbb{Q}}_{p^*}^{(i)}(x^*) - \bar{\mu}(x^*, p^*))^2}$$

and $z$ is the chosen critical value (e.g. 1.96 for a conservative bound that takes the 95% confidence interval of the bootstrap distribution). In words, the construction of $\hat{u}, \hat{l}$ equates to constructing a conservative prediction interval that depends on how dispersed or concentrated each ensemble member's predictions are.

### G.3 Metrics for Total Uncertainty Evaluation

After choosing a method to express epistemic uncertainty and combining it with aleatoric uncertainty for a prediction of total uncertainty, next comes the question of *how to evaluate the combined uncertainty*.

The critical point here is that the calibration, sharpness and proper scoring rule metrics we have discussed thus far *are not applicable here*. This is because these metric only judge how close the prediction is to the true underlying distribution. In fact, ECE (a measure of average calibration) can be shown to be identical to the Wasserstein distance between distributions under the $L1$ distance metric [65]. In a hypothetical setting where we have very few datapoints and hence very high epistemic uncertainty throughout the whole data support, if one distributional prediction was extremely lucky and predicted a distribution that adheres exactly to the true underlying distribution, the aforementioned metrics will consider this prediction a perfect prediction – however, a lucky guess is not at all a useful UQ, and to a practitioner, a less confident prediction (by quantifying high epistemic uncertainty) is much more useful, rather than a very confident prediction that can be correct if it is lucky, but confidently very incorrect otherwise.

We also emphasize here that, while standard evaluation experiments and metrics exist for the *classification* setting (e.g. by training an image classifier on the MNIST dataset and testing on the Non-MNIST dataset to assess the entropy of the predicted class probabilities or the output of a trained out-of-distribution detector), there does not exists standard experiments and metrics to evaluate epistemic uncertainty in the *regression setting*.

Therefore, we propose one evaluation metric to assess combined total uncertainty, which is *sharpness subject to sufficient coverage* and we will refer to this metric as "epistemic coverage". Epistemic coverage measures average calibration of a centered prediction interval, but the difference in observed probabilities from expected probabilities is penalized only when the observed probability is less than the expected probability (i.e. do not penalize the prediction if the observed probability is higher than the expected probability), and if this sufficient coverage condition is met, then we evaluate sharpness.

By this metric, only over-confidence is penalized and under-confidence is considered acceptable. However, since infinitely wide prediction intervals are also not useful, we also consider sharpness after sufficient coverage is met.

### G.4 Demonstrating Effect of Bootstrap Ensembling and the Epistemic Coverage Metric

We design an "epistemic experiment" to show the effect of incorporating epistemic uncertainty via bootstrapped ensembles. In this experiment, we swap the train and test sets, such that the training set is much smaller than the test set (roughly $\frac{1}{7}$ of test set size), hence, the model should have very high epistemic uncertainty in making predictions on the test set. It is expected that by not incorporating epistemic uncertainty in such a setting, the model will produce *overconfident* predictions that are

*too sharp*. This overconfidence will be penalized heavily by the epistemic coverage metric we described above in Appendix G.3, and sharpness should also indicate that the predictions are too tight. Producing conservative distributional predictions via the bootstrapped ensembling technique described above in Appendix G.2 is expected to mitigate these overconfidence issues by incorporating epistemic uncertainty.

We show the effect on one of our methods, *Interval*, because the same effect can be observed for any of the quantile methods, including the baseline algorithms. The results are shown in Figure 34.

When a conservative PI is constructed with the bootstrapped ensemble (labelled *Interval Boot-Ens*), the epistemic coverage error decreases significantly to or near zero, which is expected given that the conservative bounds only work to widen the PI, and the epistemic coverage error only penalizes PI that are not wide enough. The increase in width is also evident in the increase in sharpness with bootstrap ensembling.

Therefore, the ensembling technique does work to incorporate epistemic uncertainty from a practical standpoint by imbuing more underconfidence into the distributional predictions. Still, quantifying and evaluating epistemic uncertainty in a regression setting is an open problem, and we leave for future work developing alternative methods of quantifying epistemic uncertainty in regression.

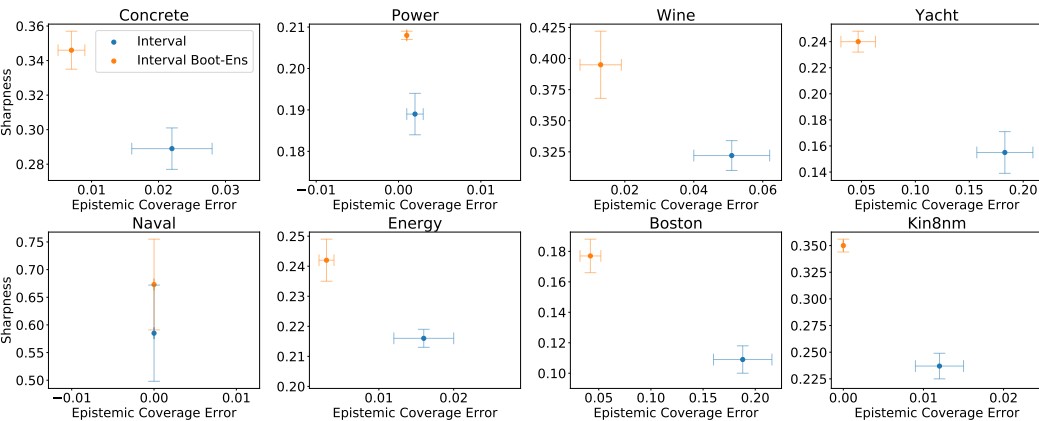

Figure 34: **Epistemic Experiments on UCI Datasets.** We evaluate epistemic coverage in the epistemic experiment setting where the train set is much smaller than the test set. Epistemic coverage only penalizes overconfidence. Sharpness is the average width of the 95% PI.

# H  Additional Discussions

**Potential negative social impacts**   This work proposes methods in uncertainty quantification (UQ), with a focus on the notion of calibration. UQ is a field that is becoming more and more important as many autonomous systems are being deployed in various real-life applications (self-driving cars, security devices, object recognition systems). While we believe the development of robust and accurate methods in UQ will accelerate deployment of autonomous systems and expand real-world use-cases, we also acknowledge the potential disruptive effects such change can have in the relevant industries.

Futher, calibration is a relevant notion in fairness [29]. Though we propose methods to achieve better calibration when making probabilistic forecasts, we note the potential for using such information with ill intent, e.g. to intentionally avoid calibrated (or fair) decisions.

**Assets used in this work**   In implementing our work, we have referenced the implementation of one of the baseline methods (*SQR*), which is publicly available under the Creative Commons Attribution-NonCommercial 4.0 International Public License.

We also state that no data from human subjects were used in this work and thus there is no personal identifying information.