# OpenReview forum: "Beyond Pinball Loss: Quantile Methods for Calibrated Uncertainty Quantification"
_NeurIPS.cc/2021/Conference — NeurIPS 2021 Poster_

### Official Review · Reviewer_58KW · 2021-07-04

**Rating:** 7
**Confidence:** 4

**Summary:**

The paper presents four alternatives to training quantile regressors with the vanilla pinball loss:

(1) a specific conditional density estimator, from which the quantiles are derived

(2) a new loss function that balance calibration and sharpness

(3) centered intervals

(4) group calibration to enforce conditional coverage

Experiments are shown to compare these methods with SQR, a neural network that optimizes the pinball loss.


**Limitations And Societal Impact:**


Yes, limitations have been addressed.


**Main Review:**

The paper is well written and the methodology is sound. In particular, I think that the new loss function (point 2) is very interesting, as well as the idea of group calibration (point 4).

The idea of deriving quantile estimators from conditional density methods (CD; point 1) has been around for some time. Thus, I miss comparisons of the conditional density estimator the authors propose to other conditional density estimators available in the literature. This is particularly relevant because, in almost all UCI datasets, MAQR was the best performing method, but it is also the only CD-based method. Thus, it is not clear to me that the good performance comes from using this specific conditional density estimator or if it is because of the fact that the problem of estimating the quantiles was being tackled in another way (namely estimating the whole distribution).

I also have two concerns with the CD method:

- The same data is being used twice: (i) to fit the regression and (ii) to model the distribution of the residuals. I wonder if regression methods that lead to overfitting (e.g., a neural network that interpolates the training points) would harm the quality of the estimator.

- The methods depend on the Euclidean distance between the features, which not good in general (especially if isotropic): It is often the case that some covariates are independent of the Y conditional on the remaining ones. This is especially true in high-dimensions. Thus, I don't agree with the authors that all nonparametric conditional densities fail in high-dims (some methods do variable selection, for instance), but instead that this specific method is not appropriate.


I also miss comparisons to other quantile estimators that don't directly target the pinball loss, such as quantile regression forests.


Minor comment:

- The results on the appendix for the Fusion datasets indicate that the Check score is minimized by SQR, so I don't really with the last statement from Section 4.1.

**Time Spent Reviewing:**

10

---

> ### Author Response · Authors · 2021-08-10
> **Response to Reviewer 58KW**
>
> Thank you very much for your review.
>
> * We agree that the idea of quantiles estimates from conditional density methods is not new; however we argue that the contribution of Point 1 is not in the specific choice of density estimator, but in utilizing this density estimator to learn a single model where quantile predictions can be made by doing a forward pass in a neural network. We are not necessarily arguing that our choice of CD is the best one, and it would be an interesting experiment to try other density estimators.
>
> * We agree that your first concern regarding MAQR is a valid one; however, we do not believe this happened in practice for us given the good test set results.  We suppose that, at the price of less data, it may be safer to split the training data into two separate sets, and it would be interesting to see how this procedure compares on the UCI benchmark. We will discuss this point in the final version of our paper.
>
> * Regarding your second comment on high dimensional settings, we appreciate you bringing up this point. It is true that the specific algorithm stated (Algorithms 1, 2) is a simple nonparametric method, which may be particularly ill-suited for high-dimensional settings. We chose this specific method to demonstrate MAQR with a more basic and standard estimator. However, using more complex conditional density estimators (that are better suited for higher dimensions) is definitely an interesting direction to explore, and we will tone down the statement made in L190 to reflect your point.
>
> * Please refer to Section F.4 and Figure 19 in the Appendix for a comparison with other quantile-based methods, including Quantile Regression Forests and Quality-Driven Prediction Intervals ([44] Pearce et al., ICML 2018).
>
> * Regarding your comment on the last statement from Section 4.1, you are correct that SQR performs the best in terms of the check score (which it optimizes directly), and we will restate our claim accordingly.
> However, we want to mention that the main case we make in this paper is that the check score (i.e. pinball loss) is inadequate for effectively assessing the performance of UQ. The fact that SQR has good check score but bad calibration provides additional evidence that the pinball loss is neither a good metric nor a good optimization objective for calibration.

---

> > ### Comment · Reviewer_58KW · 2021-09-10
> > **Thanks**
> >
> > Thank you for the helpful clarifications!

---

### Official Review · Reviewer_4TMr · 2021-07-09

**Rating:** 7
**Confidence:** 2

**Summary:**

The authors propose a new quantile method called MAQR for uncertainty quantification. This method is based on estimating empirical CDFs for residuals locally and therefore suffers from the curse of dimensionality. To address the issue, the authors further propose an NNs based solution, where the proposed loss consists of a convex combination of two terms. The first term encourages average calibration and the second term encourages sharpness. Thus allowing the user to control the balance between the two. The paper also provides a theoretical result stating that the first term is indeed minimized by the true quantile function. The authors also propose a version of the method for estimating centered intervals as well.

The paper is concluded with a rather extensive empirical evaluation.


**Main Review:**

The proposed methods are interesting and can be of interest to both researchers and practitioners interested in uncertainty quantification. The proposed methods are novel to the best of my knowledge - although I am not completely up to date with the literature.  The paper is dense, but overall well-written. However, some important definitions are not stated anywhere in the paper (see comments below). The claims of the paper are reasonably justified using a combination of theoretically motivated arguments and an extensive empirical study. The authors are comparing and evaluating the methods using several evaluation metrics. The main paper is also accompanied by a rather thorough appendix providing the details of the theoretical results as well as the experimental settings.


- It is not entirely clear to me, why eq. (6) is a good way to encourage sharpness. In my opinion, the authors should elaborate on this.

- Line 252: What is meant by: "all expected probabilities p ∈ (0, 1)" in line 252. Can the authors elaborate on that?

- What is the exact definition of sharpness used in the paper? Sharpness is both for motivating the work as well as an evaluation metric, but the authors do not provide an explicit definition.

- The MAQR method has a hyperparameter controlling bandwidth or locality, $d_N$, and one might suspect that the results can be quite sensitive to perturbations in this parameter. It would be nice if the authors would include a small discussion of the robustness of the methods.

- The objective in eq. (7) has the hyperparameter lambda balancing sharpness and calibration. It is state that lambda is estimated using cross-validation based on the adversarial group loss. I am curious to know what were the optimal values for the different experiments?

- The y-labels are missing for Figures 7 and 13 in the appendix.

# After rebuttal
I've updated my score from 6 to 7

**Time Spent Reviewing:**

2

---

> ### Author Response · Authors · 2021-08-10
> **Response to Reviewer 4TMr**
>
> Thank you very much for your review.
>
> * We follow the definition of sharpness as provided in [17] Gneiting et al., which says sharpness is purely defined by how dispersed or concentrated the predicted distribution is (L96), with more concentrated distributional predictions considered sharper and more desirable (but subject to calibration). With quantile outputs, this translates into how close together the predicted quantiles are across quantile levels $p \in (0, 1)$.
>  \
> Following this notion, existing works in uncertainty quantification $\textit{with quantiles}$ have measured sharpness with the width of prediction intervals (MPIW, mean prediction interval width), as done in [52] Tagasovska & Lopez-Paz (NeurIPS 2019), [44] Pearce et al. (ICML 2018), and [48] Salem et al. (UAI 2020). This is the same quantity as defined in Equation (6), i.e. the expected (mean) width of the $|1-2p|$ prediction interval.  \
>  \
> When we report sharpness as an evaluation metric in our experiments, in line with previous works, we report the mean width of the 95% prediction interval, as noted in L289 and L768. That being said, you bring up a good point that it would increase the clarity of our paper to give this definition explicitly, so we have updated our paper to state this definition when sharpness is first introduced.
>
>
> * In L252, “all expected probabilities $p \in (0,1)$” means that we optimize the interval score simultaneously for $p \in (0, 1)$, as denoted in $E_{\alpha \in U(0, 1)}S_{\alpha}$ in L261. The term “expected probabilities” here refers to the nominal or desired probability, as described in L76.
>
>
> * We will include the lambda values used in the Appendix of our paper, which are the following:
> |  | Concrete | Power | Wine | Yacht | Naval | Energy | Boston | Kin8nm |
> |--|---|---|---|---|---|---|---|---|
> | $\lambda$:  |  0.001 | 0.001 | 0.001 | 0.003 | 0.001 | 0.05 | 0.001 | 0.001 |
>
>
> * Thank you very much for pointing out the missing y-labels in Figures 7 and 13. We have added these labels to both Figures in our paper.
>
>
> * Regarding your comment on the hyperparameter $d_{N}$, you are correct that a wrong value for $d_N$ will lead to bad conditional density estimates, and thus affect the performance of the resulting quantile model. This would be analogous to the importance of kernel hyperparameters in Gaussian processes or kernel regression. Just as kernel hyperparameters would be tuned prior to fitting a GP or a kernel regression model, we tune $d_N$ with cross-validation with the adversarial group calibration criterion, and we find this give us a robust value which works consistently well, as shown in our empirical results.
>
>
> Finally, if you find our response satisfactory, we respectfully ask that you may consider increasing the rating. Thank you again for your time and effort in helping to polish this work.

---

> > ### Comment · Reviewer_4TMr · 2021-08-27
> > **Updating score from 6 to 7**
> >
> > Thank you for the response. You carefully addressed all of my comments, so I've updated my score from 6 to 7.

---

### Official Review · Reviewer_ePFY · 2021-07-15

**Rating:** 7
**Confidence:** 4

**Summary:**

This work is motivated by some shortcomings of the classical pinball loss used to learn quantile models, like the required differentiability of the models and miscalibration issues.
The pinball loss is also not naturally suited to build centered intervals.
This work proposes alternatives ways for learning quantile models, addressing these shortcomings.
Four methods are proposed:
- Model Agnostic QR: it is a generic method based on conditional density estimation and standard regression. Therefore it is not particularly tied to differentiable models and doesn't use the pinball loss.
- Combined calibration loss: a loss to be used instead of the pinball one, with the advantage of explicitly balancing  sharpness and calibration of the predictions.
- Centered intervals: an alternative loss based on the Winkler score is proposed to train models providing centered intervals.
- Group batching: a heuristic replacing uniform sampling for batch-based based methods, aiming at better individual calibration.

**Limitations And Societal Impact:**

The authors discussed the limitations of MAQR related to high-dimensional data.
It would be desirable to discuss limitations of the other methods (i.e. combined calibration loss, centered intervals and group batching).

The potential negative societal impact is adequately addressed in the appendix.


**Main Review:**

## General aspects

**Originality**

As far as I know, Proposition 1 is a new result that shows that the pinball loss doesn't guarantee calibration on a finite dataset.
Four methods are proposed, with different degree of originality:
- MAQR is a new quantile method leveraging existing conditional density methods like KDE and standard (mean) regression methods.
- The combined calibration loss combines two new objectives addressing sharpness and calibration respectively. A new theoretical result shows that calibration objective is well founded (it is minimized by the true quantile function).
- The centered interval  method leverages the existing interval (or Winkler) score. The novelty is in using it as training loss---instead of an evaluation metric---and in averaging it over all interval widths.
- The group batching is a heuristic that consists in using some non-uniform batching criterion with the goal of approximating adversarial group calibration that is closer to the ideal individual calibration.


**Quality**

The theoretical results are useful and the proofs seem correct.
The experiments properly illustrate the benefits of the methods proposed.

The methods are interesting. The two first methods (MAQR and combined calibration loss) are treated in more depth and rigor  than the last two. In particular, the group batching method (Section 3.4) is vaguely described (some elements are given in the appendix but still in an informal way).


I found the relationship with regularization not properly discussed. In particular, the assertion "While this may seem like an issue that can simply be addressed with regularization, we demonstrate in Appendix F.2 how that is not the case." is too strong, since appendix F.2 shows  it  experimentally for SQR only, which has the particularity of taking the quantile level as input.
A discussion of previous work relating regularization and calibration would be desirable :

*I. Takeuchi, Q. V. Le, T. D. Sears, and A. J. Smola. Nonparametric quantile estimation. JMLR, 2006.*

*I. Steinwart et al. Estimating conditional quantiles with the help of the pinball loss. Bernoulli, 2011*

It would be also interesting to compare with the following recent work that addresses the coverage problem:

*Romano, Y., Patterson, E., & Candes, E. (2019). Conformalized quantile regression. Advances in Neural Information Processing Systems, 32, 3543-3553.*


**Clarity**

The paper is well organized and well motivated. The group batching method not properly described, with some important elements being described in the appendix only. Algorithms 1 and 2 seem to contain some errors that make them confusing (see detail below).


**Significance**

The methods and the theoretical properties presented in this work should be of interest for the statistical and machine learning communities working on quantile regression and uncertainty quantification in general, and for practitioners requiring predictions with uncertainty estimation.

## Specific comments about the methods

**MAQR**

I think I got the idea from the text, but Algorithms 1,2 and 3 are confusing to me.
The idea seems to be (please correct me if I'm wrong):
- Learn a standard regression model $\hat{f}:x\rightarrow y$, compute residuals $\epsilon=y-\hat{f}(x)$
- Learn an empirical conditional CDF of the residual $\epsilon$ for each $x_k$ using its neighbors (KDE)
- Use it to compute the cumulative density $p$ of each neighbor $x$ and build a new observation $\\{x, p, \epsilon\\}$
- Learn a regression model $\hat{g}: x,p \rightarrow \epsilon$ , which can be interpreted as quantile regression model since it takes the quantile level $p$ as input.
- Use $\hat{f}+\hat{g}$

Algorithms 1 and 2:
For i=1 in Algo 1, the dataset D that is passed to Algo 2 seems to be empty.
The dataset of pairs that is taken as argument of Algo 2 seems to be the union of sets returned by itself on previous calls, but it is confusing that Algo 2 returns a set made of triplets .

Algorithm 1: why does it output k = 1, …, m ? What is m?

Line 6 : why the preceding " D: " ?   (also in Algo 2)

Maybe I am wrong because of the previous confusions, but it seems to me that the dataset that is used to train $\hat{g}$ can be larger than the original dataset (because of the overlapping neighborhoods). This should be discussed.

Algorithm 3 looks simpler and it is not obvious how Algo1+2 is an instance of it.
Line 5 of Algo 3: what does " KDE estimate of Q(xi; pk)" mean? Q is not a density.

The statement "residuals … which produces more accurate empirical CDFs" should be justified.


**Combined Calibration loss method**

The empirical loss should be introduced and explained in the main text.
It would be interesting to report the values of lambda selected by the cross-validation procedure to see what kind of tradeoff is best for the considered datasets.


**Centered Interval method**

Expectation over alpha uniform on [0,1] can make  $\hat{l}>\hat{u}$  (lower bound > upper bound). Does it make sense?

This loss also combines sharpness and calibration by taking  into account the width of the interval $u-l$ and penalizing the points outside the interval. In the light of the previous section, I would expect some discussion on how these objectives are weighted.


**Group batching**

 This is an interesting heuristic to add on previous methods and other batch based methods, by modifying how batches are built. Nevertheless, I found it not properly described. It is essentially described by the following sentence in the main text:
"Instead, deliberately grouping the datapoints based on input features, and then batching and taking
gradient steps based on these groups, induces better adversarial group calibration."
Appendix gives more information but remains very informal.

Moreover: in appendix (lines 692-694), it is said that uniformly sampled batches are alternated with the adversarial ones without justification.


## Minor comments

"Proper scoring rules" should be formally defined in Section 2.

Theorem 1: the random variable E deserves a more explicit definition.

Eq. 5 $\hat{p_p}$ seems a duplicate notation for $p^{obs}_{avg}$.

Line 215: occur -> occurs

Line 347: "While sharpness tends to increase with group batching, higher sharpness is desirable if
there is high noise in the true data distribution. "
I think this refers to the width of the 95% centered interval: higher = less sharp.
However, "higher sharpness" sounds like the opposite.

Line 592:  I found the following notation confusing :

$$\hat{p}^{obs}_{avg}(D,\tau) \text{ of } g - \tau$$

Maybe it would be clearer to define $\hat{p}^{obs}_{avg}(D,\tau)$ of $g$ and $f$ before, using different symbols.

Last part of the proof of Prop 1 (generalization to any tau ): The step from line 592 to 593 is clear since tau is part of the set {p_j} defining ECE and therefore the inequality compares  terms of each ECE.
lines 599-601: you should explain more explicitly how you get to the same inequality between ECEs when tau is not part of {p_j} .

Line 637: state -> stated
Line 810: shrink -> shrinks


## Summary of suggestions

I would be glad to increase my score if the following aspects are addressed :
- a discussion on previous work relating pinball loss, calibration and regularization is added
- the description of Algorithms 1,2,3 is improved
- the group batching method is more precisely described

## UPDATE AFTER REBUTTAL

My suggestions have been addressed. Therefore I increase my score to 7.

**Time Spent Reviewing:**

7

---

> ### Author Response · Authors · 2021-08-10
> **Response to Reviewer ePFY**
>
> Thank you very much for your detailed review.
>
> ### Comment on regularization
> * Thank you for bringing up this point. We agree that there is a crucial difference between previous works incorporating regularization with the pinball loss and this current work: previous works consider the quantile regression problem for a fixed quantile level $p$ (i.e. set a $p$, then learn $\mathbb{Q}(x)$), while we consider the problem of learning all conditional quantiles for all levels simultaneously with a single model (i.e. learn a single function $\mathbb{Q}(x,p)$).  \
>   \
> In our paper, we point out how the pinball loss is inadequate for learning $\mathbb{Q}(x,p)$, as it over-induces sharpness and drives all of the quantile predictions too close together around the datapoints $y$. At first glance, this may seem like an overfitting issue and a simple fix one could think of is regularization, which constrains the variability of the quantile across the $x$ input. Just as the reviewer mentioned, standard methods of regularization have shown to be effective in the single quantile learning setting, (Takeuchi et al., JMLR 2016; Steinwart et al., Bernoulli 2011), where we learn $\mathbb{Q}(x)$ for a fixed $p$.  \
>   \
> However, we show that this is not the case when learning $\mathbb{Q}(x,p)$, as standard methods of regularization constrain the function class not only in $x$, but also in the $p$ dimension of input. Our experiments suggest that regularization for this choice of model drives quantiles *even* closer together, and the sharpness issue of the simultaneous pinball loss is exacerbated, as shown in Appendix F.2.  \
>   \
> We agree with the reviewer that the current statements in the paper need revision, and we will append this explanation and discussion of how regularization in the standard QR setting (i.e. learning $\mathbb{Q}(x)$ for a fixed $p$) can be helpful function space constraints, but not necessarily for learning $\mathbb{Q}(x,p)$.
>
> ### Specific comments about MAQR method
> * You are correct about the error in notation for $D$ and we apologize for the confusion. The error stems from overloading $D$ for both the residual dataset $\\{x_i,\\epsilon_i\\}_{ i=1 }^{N}$, and the quantile dataset $D$ from Line 3 of Algorithm 1 (which is populated via Algorithm 2). We believe the following fix in notation should clarify the confusion:
> In Algorithm 1:
>     * Line 2: Calculate residuals $\epsilon_i, i \in [N]$ and denote the residual dataset $R=\\{x_i,\\epsilon_i\\}_{ i=1 }^{N}$.
>     * Line 5: $D_i \leftarrow$ CondQuantileEstimators$(R, i)$ (Algorithm 2)
>
>
> * Thus, Algorithm 2 always takes in a dataset of pairs $\\{x_i, \epsilon_i\\}$ and outputs a dataset of triplets $\\{x_k, \hat{p}_{k, j}, \epsilon_j\\}$, and $D$ in Algorithm 1 accumulates these datasets of triplets.
>
>
> * We apologize, please ignore $k = 1, …, m$ in Line 9 of Algorithm 1, and the preceding $D:$ in Line 6 of Algorithm 1 and Line 7 of Algorithm 2 -- we will delete these in the final version of our paper.
>
>
> * You are absolutely correct that the dataset $D$ to train $\hat{g}$ is much bigger than the original train set because of overlapping neighborhoods and because multiple quantile level estimates are made at each $x_i$. While we mention this in Appendix Section C.2 while analyzing the space complexity ($O(N^{2})$ for the quantile dataset $D$, given $N$ original training points) and in L706, we will make this point more explicit in the main body.
>
>
> * Algorithm 3 provides the general outline of MAQR, the crux of which is: (1) utilize conditional density estimators to collect a dataset of quantile estimates (Lines 3-7), and (2) learn a regression model with this dataset of quantile estimates (Line 8).
> With Algorithms 1, 2, we simply aimed to provide one practical implementation of the general MAQR algorithm.
> The specific implementation choices made for Algorithms 1, 2 are:
>     * The KDE estimator is done with a uniform kernel over $\mathcal{X}$ (Line 2, Algorithm 2: $d_N$ specifies the kernel width), and a uniform kernel over $\mathcal{Y}$ (Line 3, Algorithm 2: the empirical CDF weights all points in $E_{k, d_N}$ *uniformly*).
>     * The KDE estimator is first applied on the residuals $\{\epsilon_i\}$, and the KDE estimates of the actual targets $y_i$ are recovered by translating the residual KDE estimates by the mean function $\hat{f}$ (Line 9, Algorithm 1).
>
>
> * Please note here that taking the residuals is not a necessary step, but simply an implementation choice to demonstrate how MAQR can be readily applied in the practical setting where an accurate mean model already exists ($\hat{f}$ in Line 1, Algorithm 1).  \
>   \
> Further, existing works in conditional density estimation also suggest that having 0 conditional mean in the data provides benefits in terms of lower integrated asymptotic mean squared error (Section 5.1 and Figure 3 in [23] Hyndman et al.), which provides theoretical grounding about the benefit of working with residuals instead of the target values directly. We will mention this point in the main paper for clarity.
>
>
> * Lastly, in Line 5 of Algorithm 3, by “KDE estimate of $\mathbb{Q}(x_i, p_k)$”, we mean to say “an estimate of $\mathbb{Q}(x_i, p_k)$ derived from a conditional density estimator” (please note here that $\mathbb{Q}(x_i, p_k)$ is the true $p_k^{th}$ conditional quantile at $x = x_i$). We will clarify this line as well.
>
> ### Specific comments about Combined Calibration Loss method
> * Thank you for your suggestion about the empirical objective terms - we will update our paper to place these terms next to the expectation terms for both the calibration and sharpness objectives.
>
>
> * We will also include the $\lambda$ values used for the combined calibration loss method in the appendix, which are the following:
> |  | Concrete | Power | Wine | Yacht | Naval | Energy | Boston | Kin8nm |
> |--|---|---|---|---|---|---|---|---|
> | $\lambda$:  |  0.001 | 0.001 | 0.001 | 0.003 | 0.001 | 0.05 | 0.001 | 0.001 |
>
> ### Specific comments about Centered Interval method
>
> * We are not completely sure we understand your concern about the expectation over $\alpha$ causing $\hat{l} > \hat{u}$. Please note that the type of model we optimize the interval score with is still a "quantile model" (L68), which takes as input both an $x$ and a quantile level $p$. Therefore, when optimizing the interval score at a given $x$, for a different $\alpha$, a different lower and upper bound are predicted: $\hat{l}=\hat{\mathbb{Q}}(x, {\frac{\alpha}{2}})$, $\hat{u}=\hat{\mathbb{Q}}(x, 1-\frac{\alpha}{2})$.
> Therefore, the expectation over $\alpha \sim \text{unif}(0, 1)$ is not taken for a fixed $\hat{l}, \hat{u}$, and will not necessarily lead to the lower bounds crossing over the upper bounds.
>
>
> * You are absolutely correct that the interval score can be easily decomposed into calibration and sharpness components. In its definition, the weighting between these terms is set to $\frac{2}{\alpha}$, which are specifically the value that recovers the true quantiles at the score’s optimum, as shown in Appendix B.4. However, whether other weightings still satisfy this optimum property while improving quantile estimates is certainly an interesting direction, and we are happy to mention this point in the final version of our paper.
>
> ### Specific comments about Group Batching
> * We apologize for the vague description of group batching, and we will expand on the description in the final version of our paper. We wished to convey that the concept of group batching is quite general and allows for variations in implementations. The crux of group batching is simply to avoid drawing batches *only* from $\mathbb{F}_{\mathbf{X}}$ (the marginal distribution of $\mathbf{X}$).
>
> * For clarity, here is the exact version we implemented in our work:
>     * **Input**: Train data $D=\\{x_i, y_i\\}_{i=1}^{N}$, batch size $B$, total number of training epochs $EP$, group batching frequency hyperparameter $f$
>     * Assume $x_i \in \mathbb{R}^{m}$ and denote $x_{i, j}$ as the $j$th component of $x_i$ (i.e. $x_i = [x_{i, 1}, x_{i, 2}, \ldots, x_{i, m}]$)
>     * **for** $epoch = 1$ **to** $EP$ **do**
>         * **if** epoch%f == 0: (this is the group batching epoch)
>             * **for** $j = 1$ **to** $m$ **do**
>                 * Sort $\\{x_i\\}_{i = 1, \ldots, N}$ by values in the $j^{th}$ dimension and store this ordering of the $N$ points as $R$.
>                 * Let $R_{a:b}$ be the selection of the $a^{th}$ sorted point to the $b^{th}$ sorted point in $R$ for positive integers $a, b$.
>                 * **for** $b = 1$ **to** $\textrm{floor}(N/B)$ **do**
>                     * Do a batch update using $R_{(B*(b-1)+1) : (B*b)}$
>         * **else**: (this is the regular, uniform batching epoch)
>             * Draw random uniform batches from the original dataset $D$ and perform batch updates for one epoch.
>
> * In words, this algorithm enforces training on batches of data points that are close in a particular dimension. We do this to encourage the model to be calibrated on particular regions of input space (and therefore be closer to individual calibration). An ideal implementation of this idea would sort data by all dimensions; however, we found that the current implementation provides improvement without being too computationally demanding.
>
> * Lastly, please note that we don’t mean to say drawing batches from $\mathbb{F}\_{\mathbf{X}}$ is detrimental; rather, *only* drawing batches from $\mathbb{F}\_{\mathbf{X}}$ is, and that is why we alternate uniform batches with group batches in our specific implementation.
>
> We appreciate all of your comments and suggestions in the Minor Comments section; we will reflect the fixes and changes in the final version of the paper.
>
> Finally, if you find our response satisfactory, we respectfully ask that the reviewer considers increasing their score. If it is still unsatisfactory, please let us know if there is anything else that we can do or clarify to improve this paper..

---

> > ### Comment · Reviewer_ePFY · 2021-08-31
> > **An additional comment on the connection between Algorithms 1,2 and 3**
> >
> > Thanks for your detailed response addressing my concerns.
> >
> > Regarding the MAQR method, the algorithms are clearer with the proposed corrections.
> > I would suggest an additional clarification regarding Line 4 of algorithm 3 "select a set of quantile levels":
> > In Algorithm 2, these levels are not "selected" but indirectly given by the neighborhood E. It would be nice to clarify this in the appendix when you explain how algorithms 1&2 are an instance of 3.
> > I also suggest to replace Line 6 of Algorithm 2 by the following simpler line:
> > $$\hat{p}\_{k,j} \leftarrow \hat{\mathbb{F}}\_{E|x_k}(\epsilon_j)$$
> > I am happy to increase my score to 7.

---

### Official Review · Reviewer_ed7u · 2021-07-18

**Rating:** 7
**Confidence:** 4

**Summary:**

This paper studies procedures for quantifying uncertainty in a regression setup, focussing on the full quantile function as a form of interpretable output. The main contributions of the paper include 1) exhibiting that optimizing the pinball loss can lead to poor quantile functions, and 2) proposing new quantile methods which work in an agnostic manner.

**Limitations And Societal Impact:**

The authors provide a discussion on this and relate their work to the literature on fairness in ML.

**Main Review:**

The problem studied in the paper seems quite relevant in the context of current large-scale deep learning models being mis-calibrated, for regression as well as classification tasks.
- Overall, I enjoyed reading this paper and it is in general well-written. With the limited space though, I would urge the authors to go into a bit more detail about some of the proposed methods rather than listing all of them in Section 3.
- While the MAQR method seemed to perform well on the UCI datasets, it was not tried on the fusion dataset due to compute issues. I would be interested in seeing some simulation experiments for problems with varying samples (n) and dimensions (d). This would allow us to understand the breakdown point of the MAQR method in some more detail and provide a comparison with the other baselines.
- Since one of the key contributions of the paper is proposing new methods, I am a bit worried about the choice of the benchmark datasets from UCI. In particular, why were these particular set of 8 datasets  chosen from all the available ones? Can the authors provide evaluation on some other larger scale datasets?
- Choice of network architecture: the results were obtained with a single two-layer ReLU network. This seems quite limiting, and it would be interesting to see consistent trends across a variety of depths (even going to larger depths to showcase trends). Given the particular choice, I am not particularly convinced that the method works well in the extreme over-parameterized regime.
- Algorithm 1 (MAQR) uses residuals instead of directly working with the y values. Is there an advantage in doing so? Can you perhaps performa an ablation over these two choices to showcase why this particular form was chosen?


The authors have addressed my concerns and I have updated my rating accordingly.

**Time Spent Reviewing:**

2

---

> ### Author Response · Authors · 2021-08-10
> **Response to Reviewer ed7u**
>
> Thank you very much for your review.
>
> * Regarding your comment on the choice of benchmark UCI datasets, to the best of our knowledge, this set is standard for the regression task in uncertainty quantification, and many previous works have provided evaluations on the same datasets: [33] Lakshminarayanan et al. (NeurIPS 2017), [44] Pearce et al. (ICML 2018), [11] Detlefsen et al. (NeurIPS 2019), and our most relevant baseline SQR - [52] Tagasovska & Lopez-Paz (NeurIPS 2019) provides evaluations on the exact same set of 8 datasets. Therefore, we wanted to show results on these benchmark datasets in order to have an apples-to-apples comparison with previous methods.
>
>
> *  We believe that the fusion experiments adequately subsidize the UCI datasets since each of the 10 datasets has 468 dimensional inputs and 100k training points. That being said, we would also be happy to consider other datasets if you had suggestions. Another point to stress here is that these experiments leverage a different neural network architecture than the UCI experiments (3 layer network with 100 hidden units per layer); however, we do agree that it would be of interest to see how performance changes with network depth.
>
>
> * The reason why we use residuals in Algorithm 1, 2 is because i) existing works in conditional density estimation suggest that having 0 conditional mean in the data provides benefits in terms of lower integrated asymptotic mean squared error (Section 5.1 and Figure 3 in [23] Hyndman et al.) for the conditional density predictions, and ii) to demonstrate how MAQR is readily applicable in the practical setting where an accurate point prediction model may already exist.  \
>   \
> In the main paper, we simply aimed to provide one practical implementation of MAQR, which caters easily to the setting where a mean model already exists. However, as the general MAQR algorithm suggests (Algorithm 3 in Appendix Section C.1), the conditional density estimates can be obtained from *any* conditional density estimator for the targets $y$ (Line 5, Algorithm 3), and working with residuals is not a necessary component. We will emphasize this point in the final version of the paper.
>
>
> * With respect to the additional experiments that you have suggested (ablation on extreme over-parametrization, ablation with modeling the targets directly instead of residuals,...), we think they’re all interesting ideas, and we’ll provide updates with results as we get them.

---

> > ### Author Response · Authors · 2021-08-17
> > **We conducted the two ablation studies requested, and describe details and results below**
> >
> > ### Ablation studies on over-parameterization
> >
> > We conducted an ablation study where we significantly increased the model size and re-ran all of the methods on the Boston UCI dataset (first UCI dataset in alphabetical order). The model configurations we tested were fully connected networks that had: 5 layers with 100 hidden units per layer, 10 layers with 100 hidden units, 5 layers with 200 hidden units, and 10 layers with 200 hidden units per layer.
> >
> > As the tables below indicate, we don't see any significant deviations in the empirical results as we significantly over-parameterize the models. Please note that the training for all methods incorporates a validation set which helps to prevent overfitting by early-stopping (as mentioned in the Experiment Details in Appendix D.1, L676).
> >
> > Column labels:
> > - AvgCal: Average calibration (multiplied by 100 to follow units as reported in Figure 2)
> > - Sharp: Sharpness (multiplied by 100 to follow units as reported in Figure 2)
> > - Check: Check score
> > - Interval: Interval score
> > - IntCal: Centered interval calibration
> >
> > Note: Mean score across 5 trials is given, along with $\pm$ 1 standard error, and the best mean value for each metric has been bolded.
> >
> > > **5 layers, 100 hidden units per layer**
> >
> > | Methods    | AvgCal            | Sharp             | Check                 | Interval              | IntCal                |
> > |------------|-------------------|-------------------|-----------------------|-----------------------|-----------------------|
> > | SQR        | 10.3 $\pm$ 1.0    | **9.2** $\pm$ 1.2 | 0.086 $\pm$ 0.007     | 1.179 $\pm$ 0.130     | 0.161 $\pm$ 0.029     |
> > | mPAIC      | 8.6 $\pm$ 2.0     | 13.2 $\pm$ 1.1    | 0.093 $\pm$ 0.007     | 1.382 $\pm$ 0.208     | **0.057** $\pm$ 0.009 |
> > | *Interval* | 6.9 $\pm$ 0.9     | 17.3 $\pm$ 1.3    | 0.093 $\pm$ 0.008     | 1.093 $\pm$ 0.169     | 0.085 $\pm$ 0.021     |
> > | *Cali*     | 8.4 $\pm$ 1.0     | 11.7 $\pm$ 0.5    | 0.097 $\pm$ 0.009     | 1.310 $\pm$ 0.180     | 0.129 $\pm$ 0.017     |
> > | *MAQR*     | **6.0** $\pm$ 2.3 | 10.5 $\pm$ 0.8    | **0.062** $\pm$ 0.016 | **0.861** $\pm$ 0.310 | 0.085 $\pm$ 0.045     |
> >
> > > **10 layers, 100 hidden units per layer**
> >
> > | Methods    | AvgCal            | Sharp             | Check                 | Interval              | IntCal                |
> > |------------|-------------------|-------------------|-----------------------|-----------------------|-----------------------|
> > | SQR        | 8.8 $\pm$ 1.4     | 10.8 $\pm$ 1.3    | 0.088 $\pm$ 0.007     | 1.170 $\pm$ 0.127     | 0.131 $\pm$ 0.029     |
> > | mPAIC      | **5.1** $\pm$ 1.0 | 14.7 $\pm$ 1.6    | 0.097 $\pm$ 0.006     | 1.399 $\pm$ 0.128     | **0.064** $\pm$ 0.019 |
> > | *Interval* | 5.9 $\pm$ 0.9     | 17.8 $\pm$ 0.8    | 0.086 $\pm$ 0.007     | 0.950 $\pm$ 0.108     | 0.075 $\pm$ 0.014     |
> > | *Cali*     | 6.8 $\pm$ 0.8     | 13.2 $\pm$ 0.4    | 0.099 $\pm$ 0.011     | 1.272 $\pm$ 0.200     | 0.105 $\pm$ 0.013     |
> > | *MAQR*     | 6.6 $\pm$ 2.1     | **9.9** $\pm$ 0.8 | **0.062** $\pm$ 0.016 | **0.850** $\pm$ 0.300 | 0.092 $\pm$ 0.042     |
> >
> > > **5 layers, 200 hidden units per layer**
> >
> > | Methods    | AvgCal            | Sharp             | Check                 | Interval              | IntCal                |
> > |------------|-------------------|-------------------|-----------------------|-----------------------|-----------------------|
> > | SQR        | 10.2 $\pm$ 1.3    | **9.8** $\pm$ 0.9 | 0.089 $\pm$ 0.005     | 1.193 $\pm$ 0.097     | 0.170 $\pm$ 0.011     |
> > | mPAIC      | 9.6 $\pm$ 1.4     | 15.8 $\pm$ 2.3    | 0.097 $\pm$ 0.005     | 1.440 $\pm$ 0.066     | 0.092 $\pm$ 0.028     |
> > | *Interval* | 6.7 $\pm$ 1.0     | 19.3 $\pm$ 1.3    | 0.089 $\pm$ 0.006     | 0.983 $\pm$ 0.102     | **0.083** $\pm$ 0.015 |
> > | *Cali*     | 7.0 $\pm$ 1.0     | 12.7 $\pm$ 0.9    | 0.090 $\pm$ 0.009     | 1.119 $\pm$ 0.157     | 0.120 $\pm$ 0.018     |
> > | *MAQR*     | **6.5** $\pm$ 2.1 | 10.3 $\pm$ 0.8    | **0.062** $\pm$ 0.016 | **0.856** $\pm$ 0.299 | 0.097 $\pm$ 0.043     |
> >
> > > **10 layers, 200 hidden units per layer**
> >
> > | Methods    | AvgCal            | Sharp              | Check                 | Interval              | IntCal                |
> > |------------|-------------------|--------------------|-----------------------|-----------------------|-----------------------|
> > | SQR        | 8.3 $\pm$ 1.2     | 10.8 $\pm$ 1.3     | 0.086 $\pm$ 0.005     | 1.119 $\pm$ 0.118     | 0.138 $\pm$ 0.029     |
> > | mPAIC      | 6.6 $\pm$ 0.9     | 16.4 $\pm$ 1.4     | 0.089 $\pm$ 0.004     | 1.313 $\pm$ 0.077     | 0.095 $\pm$ 0.016     |
> > | *Interval* | 6.7 $\pm$ 1.0     | 19.3 $\pm$ 1.3     | 0.089 $\pm$ 0.006     | 0.983 $\pm$ 0.102     | **0.083** $\pm$ 0.015 |
> > | *Cali*     | 6.6 $\pm$ 1.0     | 13.9 $\pm$ 0.6     | 0.097 $\pm$ 0.009     | 1.224 $\pm$ 0.155     | 0.094 $\pm$ 0.017     |
> > | *MAQR*     | **6.1** $\pm$ 2.2 | **10.3** $\pm$ 0.7 | **0.061** $\pm$ 0.016 | **0.839** $\pm$ 0.298 | 0.087 $\pm$ 0.044     |
> >
> > \
> > &nbsp;
> >
> > ### Ablation studies on not modeling residuals for MAQR
> >
> > We also ran MAQR in the setting where we model the target $y$ values directly (instead of using residuals), on all 8 UCI datasets.
> >
> > As expected, modeling the residuals significantly helps MAQR’s performance, which declines in the alternative formulation that discards the point prediction model, as the results show below. This is simply a consequence of *worse* conditional density estimates: in addition to the theoretical statements provided by [23] Hyndman et al. (see bullet point 3 in first response above), intuitively speaking, the point prediction model essentially decomposes signal (mean) from the noise (uncertainty), thus modeling the residuals allows a more accurate description of the underlying uncertainty.
> >
> > In practice, the quality of the conditional density estimates would be examined before training the quantile model. Comparing conditional density estimators is another field of its own and somewhat orthogonal to this work, and in practice, if one estimator is superior over another, there wouldn’t be a reason to use the latter.
> >
> > > **MAQR that models the targets $y$ directly**
> >
> > | Dataset  | AvgCal        | Sharp          | Check             | Interval          | IntCal            |
> > |----------|---------------|----------------|-------------------|-------------------|-------------------|
> > | concrete | 5.7 $\pm$ 1.1 | 42.4 $\pm$ 2.8 | 0.169 $\pm$ 0.007 | 1.694 $\pm$ 0.077 | 0.053 $\pm$ 0.010 |
> > | power    | 3.1 $\pm$ 0.2 | 24.0 $\pm$ 0.2 | 0.065 $\pm$ 0.001 | 0.653 $\pm$ 0.008 | 0.050 $\pm$ 0.003 |
> > | wine     | 4.6 $\pm$ 0.7 | 29.8 $\pm$ 2.3 | 0.213 $\pm$ 0.007 | 2.571 $\pm$ 0.140 | 0.059 $\pm$ 0.010 |
> > | yacht    | 9.8 $\pm$ 0.9 | 45.8 $\pm$ 2.3 | 0.124 $\pm$ 0.009 | 1.298 $\pm$ 0.106 | 0.185 $\pm$ 0.020 |
> > | naval    | 3.2 $\pm$ 0.1 | 81.3 $\pm$ 0.3 | 0.203 $\pm$ 0.002 | 1.910 $\pm$ 0.016 | 0.058 $\pm$ 0.003 |
> > | energy   | 4.9 $\pm$ 0.4 | 29.0 $\pm$ 0.8 | 0.068 $\pm$ 0.002 | 0.656 $\pm$ 0.018 | 0.083 $\pm$ 0.002 |
> > | boston   | 3.8 $\pm$ 0.4 | 31.8 $\pm$ 1.5 | 0.132 $\pm$ 0.007 | 1.550 $\pm$ 0.092 | 0.054 $\pm$ 0.005 |
> > | kin8nm   | 6.9 $\pm$ 0.2 | 49.4 $\pm$ 0.1 | 0.134 $\pm$ 0.001 | 1.384 $\pm$ 0.005 | 0.135 $\pm$ 0.003 |

---

### Decision · Program_Chairs · 2021-09-27

**Decision:**

Accept (Poster)

**Comment:**

In this study, the authors are concerned with the problem of estimating the full quantile function of the responses in a regression setting. This task can be typically realized by performing quantile regression on all quantiles. However, the authors pointed out a drawback of pinball loss, which is usually used in quantile regression, and a new method that can overcome this drawback is proposed. Both the problem and the setting discussed in this paper are important, and the theoretical validity of the proposed method and the demonstration of its performance through experiments are properly done. All reviewers found the paper useful for the ML community. The authors are encouraged to consider updating the paper based on the reviewers' comments and suggestions.